



# Retrieval improvements for the ALADIN Airborne Demonstrator in support of the Aeolus wind product validation

Oliver Lux[1], Christian Lemmerz[1], Fabian Weiler[1], Uwe Marksteiner[1], Benjamin Witschas[1], Stephan Rahm[1], Alexander Geiß[2], Andreas Schäfler[1], Oliver Reitebuch[1]

[1] German Aerospace Center (Deutsches Zentrum für Luft- und Raumfahrt e.V., DLR), Institute of Atmospheric Physics, Oberpfaffenhofen 82234, Germany
[2] Ludwig-Maximilians-University Munich, Meteorological Institute, 80333 Munich, Germany

*Correspondence to*: Oliver Lux (oliver.lux@dlr.de)

**Abstract.** The realization of the European Space Agency's Aeolus mission was supported by the long-standing development and field deployment of the ALADIN Airborne Demonstrator (A2D) which, since the launch of the Aeolus satellite in 2018, has been serving as a key instrument for the validation of the Atmospheric LAser Doppler INstrument (ALADIN), the first-ever Doppler wind lidar (DWL) in space. However, the validation capabilities of the A2D are compromised by deficiencies of the dual-channel receiver which, like its spaceborne counterpart, consists of a Rayleigh and a complementary Mie spectrometer for sensing the wind speed from both molecular and particulate backscatter signals, respectively. Whereas the accuracy and precision of the Rayleigh channel is limited by the spectrometer's high alignment sensitivity, especially in the near field of the instrument, large systematic Mie wind errors are caused by aberrations of the interferometer in combination with the temporal overlap of adjacent range gates during signal readout. The two error sources are mitigated by modifications of the A2D wind retrieval algorithm. A novel quality control scheme was implemented which ensures that only backscatter return signals within a small angular range are further processed. Moreover, Mie wind results with large bias of opposing sign in adjacent range bins are vertically averaged. The resulting improvement of the A2D performance was evaluated in the context of two Aeolus airborne validation campaigns that were conducted between May and September 2019. Comparison of the A2D wind data against a high-accuracy, coherent Doppler wind lidar that was deployed in parallel on-board the same aircraft shows that the retrieval refinements considerably decrease the random errors of the A2D line-of-sight (LOS) Rayleigh and Mie winds from about 2.0 m·s⁻¹ to about 1.5 m·s⁻¹, demonstrating the capability of such a direct detection DWL. Moreover, the measurement range of the Rayleigh channel could be largely extended by up to 2 km in the instrument's near field close to the aircraft. The Rayleigh and Mie systematic errors are below 0.5 m·s⁻¹ (LOS), hence allowing for an accurate assessment of the Aeolus wind errors during the September campaign. The latter revealed different biases of the L2B Rayleigh-clear and Mie-cloudy horizontal LOS (HLOS) for ascending and descending orbits as well as random errors of about 3 m·s⁻¹ (HLOS) for the Mie and close to 6 m·s⁻¹ (HLOS) for the Rayleigh winds, respectively. In addition to the Aeolus error evaluation, the present study discusses the applicability of the developed A2D algorithm modifications to the Aeolus processor, thereby offering prospects for improving the Aeolus wind data quality.



# 1 Introduction

Doppler wind lidar (DWL) represents a powerful means to acquire atmospheric wind profiles with high spatiotemporal
resolution and high accuracy, and it has been employed from ground as well as from air- and shipborne platforms for many
years. Following the launch of the European Space Agency's (ESA) Aeolus mission in August 2018, the measurement of
vertically-resolved wind profiles on a global scale is being accomplished by the Atmospheric LAser Doppler INstru-ment
(ALADIN), the first DWL in space. ALADIN is capable of observing the component of the wind vector along the instrument's
line-of-sight (LOS) from ground up to 30 km in the stratosphere (ESA, 2008; Reitebuch, 2012; Kanitz et al., 2019; Parrinello
et al., 2021). The Aeolus mission primarily aims at improving numerical weather prediction (NWP) by filling observational
gaps in the global wind data coverage, particularly over the oceans, poles, tropics, and the Southern Hemisphere (Andersson,
2018; Stoffelen et al., 2005, 2020; Straume et al., 2020). This goal was already achieved within the first half of the intended
mission lifetime of three years with the successful assimilation of Aeolus winds into NWP models, after having identified and
corrected for two large error sources that had diminished the wind data quality in the early phase of the mission (Kanitz et al.,
2020; Reitebuch et al., 2020). Whereas the implementation of a dedicated calibration instrument mode allows to account for
dark current signal fluctuations on the Aeolus detectors (Weiler et al., 2020), wind biases that were caused by variations in the
temperature distribution across the primary telescope mirror were considerably reduced by a correction scheme based on
ECMWF model-equivalent winds (Rennie and Isaksen, 2020; Weiler et al, 2021; Rennie et al., 2021). Recent assessments of
the significance of the Aeolus data have demonstrated statistically positive impact on the accuracy of NWP forecasts, especially
in the tropics and at the poles, hence providing a useful contribution to the Global Observing System (Rennie and Isaksen,
2020; Rennie et al., 2021). Consequently, the operational assimilation of Aeolus observations was started at several weather
services worldwide in 2020.

Since the start of the mission, the quality of the Aeolus wind data product has been assessed by a number of calibration and
validation (Cal/Val) activities based on model comparisons (Martin et al., 2021) and ground-based or suborbital instruments
(Bedka et al., 2021; Iwai et al., 2021; Fehr et al., 2020, 2021; Baars et al., 2020; Khaykin et al., 2020). Within the framework
of this international effort, the German Aerospace Center (Deutsches Zentrum für Luft- und Raumfahrt, DLR) has conducted
three airborne validation campaigns between November 2018 and October 2019 (Lux et al., 2020a), deploying its high-
accuracy, coherent 2-μm DWL (Weissmann et al., 2005; Witschas et al., 2017; Witschas et al., 2020) in combination with the
ALADIN Airborne Demonstrator (A2D) on-board the DLR Falcon research aircraft. Being the airborne prototype of the
Aeolus payload, the A2D has a very similar design and relies on the same measurement principle (Paffrath et al., 2009;
Reitebuch et al., 2009), thus delivering valuable information on potential error sources as well as on the optimization of the
Aeolus wind retrieval and related quality-control algorithms. In particular, the A2D receiver is, like ALADIN, composed of
complementary Rayleigh and Mie channels for deriving the wind speed from both molecular and particulate backscatter.
Beyond the airborne campaigns, the A2D has been serving as testbed to explore new measurement strategies and algorithm
modifications which cannot be readily implemented in the Aeolus operation modes and wind processing, respectively.



During earlier campaigns, the (pre-launch) validation capabilities of the A2D were impaired by two major deficiencies that limited the accuracy and precision of the two receiver channels. While the Rayleigh wind results are impaired by the imperfect transmit-receive co-alignment in combination with the incomplete telescope overlap in the instrument's near field, large systematic Mie wind errors are caused by imperfections of the Fizeau interferometer (FI) plates which results in a skewed

interference fringe. These two error sources were tackled by several modifications of A2D wind retrieval which considerably improved the accuracy and precision of the wind measurement while increasing the measurement range. The identified strategies to overcome the limitations imposed by the two error sources are not only specific to the A2D, but are potentially also relevant for other DWLs that utilize either FPIs or FIs for deriving Doppler frequency shifts. This is especially true for the A2D's spaceborne counterpart ALADIN, despite its monostatic design where the transmit and receive path are realized by

the same telescope which makes an active co-alignment, as used in the A2D, obsolete. However, the developed retrieval modifications are not applicable to DWLs that rely on other measurement principles, such as Mach–Zehnder interferometers (Baidar et al., 2018; Tucker et al., 2018; Bruneau and Pelon, 2021), particularly as they are generally less sensitive to angular variations.

The refinement of the A2D wind retrieval and resulting improvement of the wind data quality substantially enhanced the

capabilities of the airborne demonstrator for the Aeolus validation. Furthermore, the lessons learned from the first validation campaign in 2018 regarding the optimization of the A2D range gate settings (Lux et al., 2020a) were applied in order to increase the statistics of the Aeolus-to-A2D wind comparisons. The statistical significance is further strengthened by a larger number of underflights performed during the two airborne campaigns in 2019 that are discussed in the present work compared to the campaign in 2018 where only a small dataset was analyzed, hence providing a much higher validity of the Aeolus data

quality assessment. Finally, a major difference to the previous work is the fact that the Aeolus wind data from the most recent campaign was already reprocessed after implementation of several crucial correction schemes to the Aeolus processors for considerably reducing large systematic errors that were identified in the initial phase of the mission (Abdalla et al., 2020; Masoumzadeh et al., 2020). Analysis of the satellite data from the latest airborne campaign thus allows for the validation of the reprocessed Aeolus wind product with improved data quality.

This paper aims to elaborate on the A2D wind measurement error sources and how they were mitigated in the context of two Aeolus validation campaigns in 2019. After a brief overview of the campaigns and obtained datasets (Sect. 2), the technical evolution of the A2D is outlined, followed by a description of the A2D wind retrieval algorithm (Sect. 3.1). The latter forms the basis for the developed retrieval modifications of the Rayleigh and Mie wind retrieval which are explained in Sect. 3.2 and Sect. 3.3, respectively. The improvement in the A2D wind data quality is demonstrated by a statistical comparison with

collocated 2-µm DWL wind measurements carried out during the two airborne campaigns (Sect. 3.4.). Section 4 then focuses on the validation of reprocessed Aeolus wind results from the second campaign in 2019 using the improved A2D wind data. The article closes with a summary and conclusions drawn from the analyses (Sect. 5), including a discussion on the possible applicability of the developed correction schemes to the Aeolus wind retrieval.



## 2 Overview of validation campaigns and datasets

The airborne validation of the Aeolus mission started already three months after the launch of the satellite on 22 August 2018 when the WindVal III campaign was conducted from the DLR airbase in Oberpfaffenhofen, Germany between 5 November and 5 December 2018. In addition to performing collocated wind measurements of ALADIN and the two wind lidars on-board the DLR Falcon research aircraft, WindVal III also aimed at rehearsing the validation activities to be carried out after the commissioning phase of the mission. Balancing the constraints in the planning of the satellite underflights which were imposed

by the weather conditions along the measurement track within the reach of the aircraft, air traffic control (ATC) limitations and the satellite status was an important experience gained for the subsequent campaigns. Moreover, several lessons learned were derived in terms of the measurement strategies and settings of the A2D. Most importantly, following the recommendations formulated in Lux et al. (2020a), the range gate settings of the A2D were optimized such that a higher number of small and medium-sized range gates were located at altitudes above 4 km at the expense of lower resolution towards

the ground. The higher resolution at higher altitudes allowed for a better vertical sampling of high wind speed gradients, e.g., related to jet streams near the tropopause, and hence delivered wind data over a wider wind speed range to be used for the validation of the Aeolus wind product.

Following WindVal III in late 2018, two additional airborne validation campaigns were conducted within the first year of the operational phase of Aeolus. Whereas the AVATARE campaign (Airborne VAlidation Through Airborne LidaRs in Europe)

was again carried out from Oberpfaffenhofen, Germany in May/June 2019, AVATARI (Airborne VAlidation Through Airborne LidaRs in Iceland), performed in September 2019, had its airbase in Keflavík, Iceland which was also the location of the WindVal II campaign in 2016 prior to launch (Lux et al, 2018). The two campaigns not only considerably extended the available dataset to validate Aeolus wind observations in different geographical regions and under different atmospheric conditions in terms of cloud types and dynamics, but also served to assess the accuracy and precision of the Aeolus wind

product in different phases of the mission. In particular, the switch-over from the first flight model laser (FM-A) to the redundant laser FM-B just after AVATARE in June 2019 marked a caesura, as the FM-B delivered much higher emit energy and hence ensured higher signal-to-noise ratio of the backscatter return compared to the end of the FM-A period (Lux et al. 2020b; Lux et al., 2021). Moreover, the satellite instrument was found to be more susceptible to changes in the thermal environment than expected before launch showing orbital and seasonal variations in the beam alignment, and hence in the

wind bias (Martin et al., 2021; Chen et al., 2021). This issue, together with the influence of the seasonally varying solar background being incident on the instrument spectrometers and affecting the wind measurement precision, could be investigated based on the comparative wind measurements from the AVATARE and AVATARI campaigns.

### 2.1 AVATARE

In the framework of the AVATARE campaign, eight research flights were conducted from which six flights were dedicated to

Aeolus underpasses and two flights aimed at the calibration of the A2D. The purpose and procedure of the latter are extensively





described in Marksteiner et al. (2018). The flight tracks of the DLR Falcon are depicted in Fig. 1, while Table 1 provides an overview of the performed flights including the geolocations of the start and end points of the respective Aeolus underflight legs as well as the number of A2D and Aeolus observations, respectively.

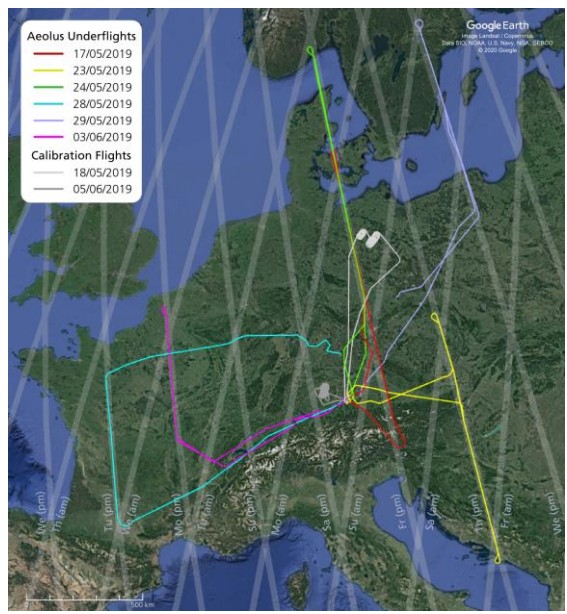


**Figure 1.** Flight tracks of the Falcon aircraft during the AVATARE campaign from 17 May to 5 June 2019 (background image: © 2020 Google). Each colour represents a single flight. The Aeolus measurement swath is shown in grey. The satellite flying direction was always from south to north during the probed evening ascending orbits.

**Table 1.** Overview of the research flights of the Falcon aircraft in the frame of the AVATARE campaign in May/June 2019 and the wind
scenes performed with the A2D along the Aeolus measurement track.

| Flight # | Date | Flight period (UTC) | Measurement period (UTC) | Number of A2D observations | Geolocation of DLR Falcon on Aeolus measurement track (start/stop) | | Number of Aeolus observations |
|---|---|---|---|---|---|---|---|
| 1 | 17/05/2019 | 15:36 – 18:46 | 16:13 – 17:13 | 200 | 47.9°N, 13.3°E | 54.7°N, 11.1°E | 10 |
| | | | 17:24 – 17:58 | 114 | 55.6°N, 10.9°E | 52.0°N, 12.0°E | 7 |
| 2 | 18/05/2019 | 14:51 – 18:31 | | | Calibration flight | | |
| 3 | 23/05/2019 | 14:30 – 18:08 | 15:13 – 15:54 | 137 | 47.7°N, 16.5°E | 42.9°N, 17.6°E | 8 |
| | | | 16:00 – 17:11 | 234 | 43.3°N, 17.5°E | 50.5°N, 15.7°E | 11 |
| 4 | 24/05/2019 | 15:28 – 19:09 | 16:04 – 17:18 | 243 | 51.2°N, 12.3°E | 58.9°N, 9.5°E | 13 |
| | | | 17:25 – 17:52 | 92 | 58.4°N, 9.8°E | 55.1°N, 11.0°E | 7 |
| 5 | 28/05/2019 | 15:54 – 19:13 | 17:17 – 17:49 | 107 | 48.3°N, 0.1°E | 44.0°N, 1.2°E | 7 |
| 6 | 29/05/2019 | 15:26 – 19:11 | 16:02 – 16:55 | 174 | 53.5°N, 18.2°E | 59.5°N, 16.1°E | 9 |
| | | | 17:05 – 17:58 | 175 | 59.2°N, 16.1°E | 53.7°N, 18.0°E | 9 |
| 7 | 03/06/2019 | 15:26 – 18:46 | 16:31 – 17:06 | 117 | 46.9°N, 3.6°E | 50.6°N, 2.6°E | 7 |
| | | | 17:11 – 17:33 | 74 | 50.3°N, 2.7°E | 47.8°N, 3.4°E | 6 |
| 8 | 05/06/2019 | 07:11 – 10:44 | | | Calibration flight | | |





As can be seen from the flight tracks and times, all underflights were in the evening hours during ascending orbits of the satellite. Descending orbits in the morning hours were not probed due to unfavorable weather conditions and/or ATC constraints during the campaign period. Adding up the lengths of the satellite swaths covered by the DLR Falcon during the six underflights, the overall track length for which wind data is available for validation purposes is around 4400 km,

corresponding to 57 Aeolus wind observations. Since parts of the swaths were overflown two times during some of the flights (see Table 1), several Aeolus observations could be validated by two collocated wind measurements of the airborne wind lidars.

## 2.2 AVATARI

Thanks to a larger number of available flight hours during AVATARI (62 compared to 34 during AVATARE), a total of ten

Aeolus underflights were performed in September 2019, including also, for the first time, four flights along descending orbits in the early morning hours (see Fig. 2 and Table 2). Already during the transfer from Oberpfaffenhofen to Keflavík via Prestwick, United Kingdom, the first underflight along an ascending orbit could be performed north of Scotland. Since there were no cloud-free conditions over the Iceland main glacier, two double flights were flown to Greenland with a fuel stop in Kangerlussuaq for the purpose of calibrating the A2D over high-albedo surfaces. During the ten underflights the two lidars

on-board the DLR Falcon sampled almost 8000 km of the Aeolus measurement track across Iceland and parts of the North Atlantic. In contrast to the previous validation campaigns, Aeolus operated with a dedicated range gate setting that was applied exclusively in the area around Iceland within the AVATARI timeframe. The setting was optimized for a higher resolution (500 m instead of 1000 m) throughout the troposphere, particularly in the jet-stream area, in order to better resolve strong wind speed gradients at the expense of increased noise.

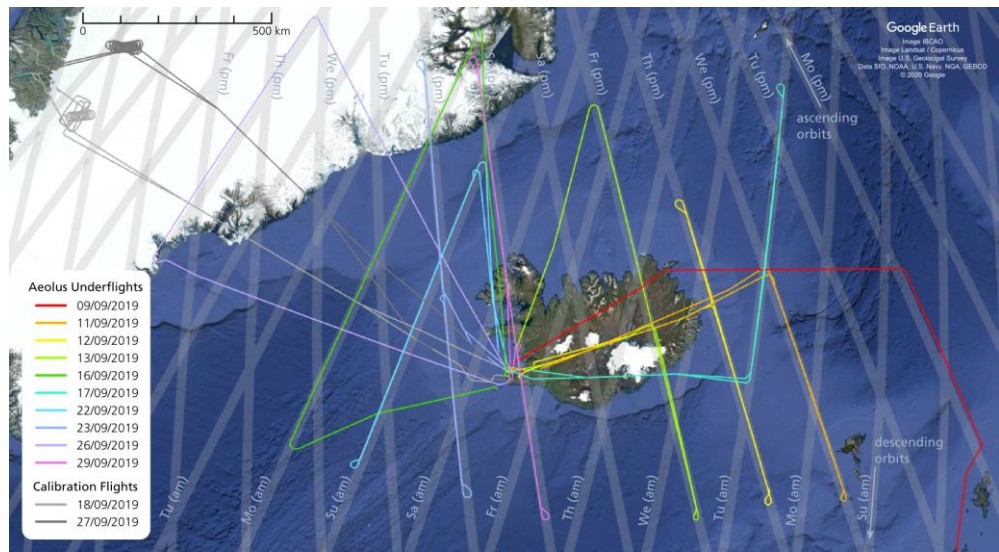


**Figure 2.** Flight tracks of the Falcon aircraft during the AVATARI campaign from 9 September to 29 September 2019 (background image: © 2020 Google). Each colour represents a single satellite underflight. The Aeolus measurement swath is shown in grey.





**Table 2.** Overview of the research flights of the Falcon aircraft in the frame of the AVATARI campaign in September 2019 and the wind scenes performed with the A2D along the Aeolus measurement track.

| Flight # | Date | Flight period (UTC) | Measurement period (UTC) | Number of A2D observations | Geolocation of DLR Falcon on Aeolus measurement track (start/stop) | | Number of Aeolus observations |
|---|---|---|---|---|---|---|---|
| 1 | 09/09/2019 | 16:12 – 19:22 | 17:17 – 17:52 | 118 | 61.6°N, 0.9°W | 65.5°N, 2.9°W | 7 |
| 2 | 11/09/2019 | 16:04 – 19:28 | 17:10 – 17:40 | 100 | 64.6°N, 9.2°W | 61.2°N, 7.5°W | 6 |
| | | | 17:45 – 18:26 | 139 | 61.4°N, 7.6°W | 66.0°N, 9.9°W | 7 |
| 3 | 12/09/2019 | 16:25 – 19:52 | 17:09 – 17:44 | 116 | 65.2°N, 12.8°W | 61.4°N, 10.8°W | 6 |
| | | | 17:49 – 18:42 | 172 | 61.6°N, 11.0°W | 67.5°N, 14.1°W | 9 |
| 4 | 13/09/2019 | 16:52 – 20:19 | 17:23 – 17:58 | 119 | 65.0°N, 15.7°W | 61.1°N, 14.0°W | 8 |
| | | | 18:02 – 19:19 | 250 | 61.3°N, 14.1°W | 69.6°N, 18.6°W | 13 |
| 5 | 16/09/2019 | 06:45 – 10:09 | 07:56 – 09:19 | 274 | 70.7°N, 26.4°W | 62.0°N, 31.8°W | 15 |
| 6 | 17/09/2019 | 05:07 – 08:43 | 05:50 – 06:43 | 174 | 64.1°N, 11.3°W | 69.7°N, 7.7°W | 10 |
| | | | 06:48 – 07:37 | 163 | 69.5°N, 7.9°W | 63.9°N, 11.4°W | 10 |
| 7 | 18/09/2019 | 08:53 – 12:30 | | | Calibration flight | | |
| 8 | 18/09/2019 | 13:38 – 16:08 | | | Calibration flight | | |
| 9 | 22/09/2019 | 06:58 – 10:30 | 08:07 – 08:37 | 99 | 65.1°N, 27.0°W | 62.0°N, 28.6°W | 7 |
| | | | 08:45 – 09:36 | 168 | 62.1°N, 28.6°W | 68.0°N, 25.2°W | 11 |
| 10 | 23/09/2019 | 17:38 – 21:18 | 18:12 – 18:40 | 94 | 64.5°N, 25.5°W | 61.9°N, 24.2°W | 5 |
| | | | 18:53 – 19:57 | 214 | 62.3°N, 24.4°W | 69.9°N, 29.0°W | 11 |
| | | | 20:08 – 20:44 | 118 | 69.3°N, 28.5°W | 65.7°N, 26.2°W | 7 |
| 11 | 26/09/2019 | 07:36 – 11:03 | 08:54 – 09:40 | 154 | 70.3°N, 36.6°W | 64.7°N, 40.4°W | 11 |
| 12 | 27/09/2019 | 09:01 – 12:25 | | | Calibration flight | | |
| 13 | 27/09/2019 | 13:54 – 17:16 | | | Calibration flight | | |
| 14 | 29/09/2019 | 17:28 – 20:47 | 18:22 – 19:40 | 259 | 61.8°N, 20.9°W | 70.3°N, 25.9°W | 13 |
| | | | 19:56 – 20:16 | 66 | 68.7°N, 24.8°W | 66.5°N, 23.3°W | 5 |

## 3 The A2D and its evolution since 2005


The A2D was developed by European Aeronautic Defence and Space (EADS)-Astrium (now Airbus Defence and Space) in cooperation with the DLR in the early 2000s. Its design and operating principle were extensively described in several previous publications (Paffrath et al., 2009; Reitebuch et al., 2009; Lux et al.; 2018, Lux et al., 2020a). Therefore, only a brief description is provided here. The A2D consists of a pulsed, frequency-stabilized, ultra-violet (UV) laser transmitter, a Cassegrain-type telescope, a front optics configuration for separating the backscatter return from a fibre-coupled internal reference signal and two complementary receiver channels that are sensitive for molecular (Rayleigh channel) and particulate backscatter from clouds and aerosols (Mie channel), respectively (Fig. 3). The structural design of the telescope, which is composed of a 20 cm diameter primary mirror and a 7 cm diameter secondary mirror, involves a strong attenuation of the return signal within the first 2 km from the instrument (Paffrath et al., 2009). Moreover, the incomplete telescope overlap in the instrument's near field



causes a range-dependent angular distribution of the return signal that is incident on the spectrometers of the receiver, hence
giving rise to systematic errors, particularly of the Rayleigh channel (see Sect. 3.2). The incomplete overlap between the
transmitted laser beam and the field of view (FOV) of the receiving telescope is an issue that impairs the near field performance
of many different types of lidar instruments and various approaches have been developed over last decades to determine the
overlap function which particularly crucial for aerosol remote sensing (Wandinger and Ansmann, 2002; Stelmaszczyk et al.,
2005; Mei et al., 2020). For the wind retrieval of the A2D, exact knowledge the overlap function is not necessary; however,
systematic errors arise from the fact that the beam obscuration by the telescope secondary mirror reduces the backscatter such
that only higher field angles are transmitted.

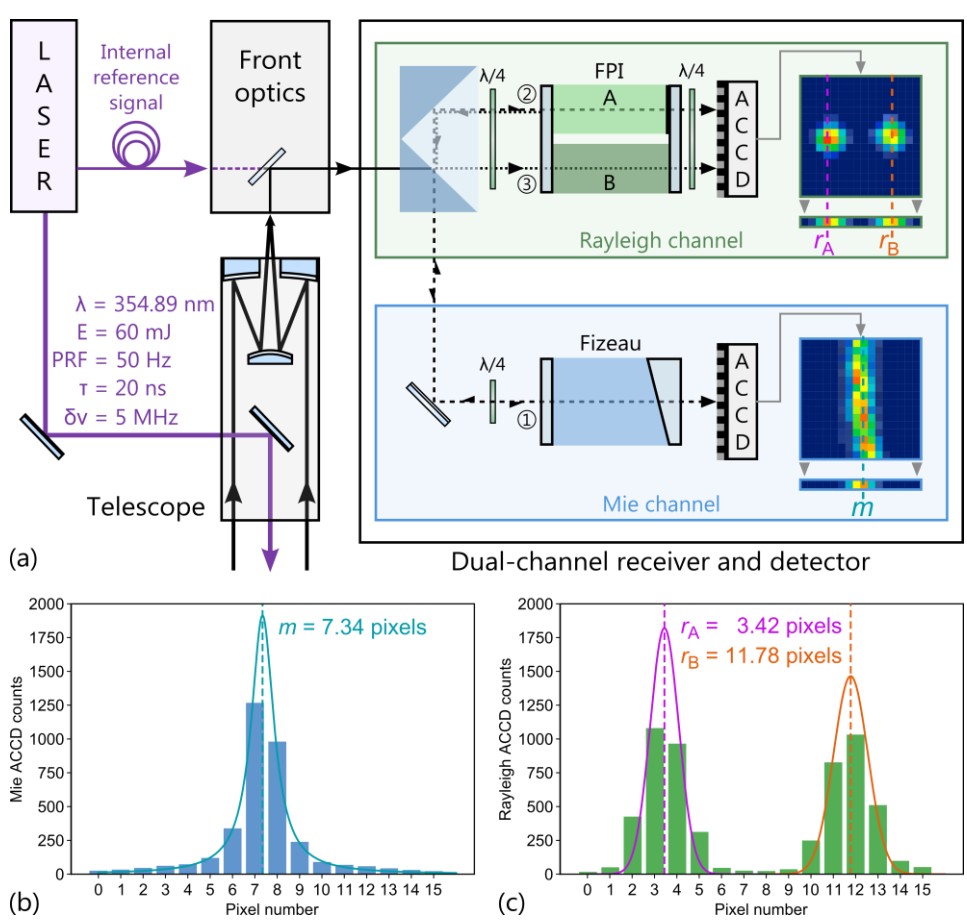

**Figure 3.** (a) Simplified schematic of the A2D comprising a frequency-stabilized, UV laser transmitter, a Cassegrain telescope, front optics
and a dual-channel receiver. FPI: Fabry–Pérot interferometer; PRF pulse repetition frequency; ACCD: accumulation charge-coupled device.
(b) Exemplary signal distribution measured with the Mie ACCD detector (blue bars) and Lorentzian fit of the signals transmitted through
the FI for determining the Mie fringe centroid position $m$. (c) Exemplary signal distribution measured with the Rayleigh ACCD detector
(green bars) and Gaussian fits of the signals transmitted through the two FPIs (A: pink, B: orange) for determining the Rayleigh spot positions
$r_A$ and $r_B$.





The wind measurement principle of the two receiver channels relies on different techniques to account for the diverse spectral bandwidth of the particulate (~50 MHz) and molecular backscatter (~3.8 GHz in the UV spectral region and at atmospheric temperature of 293 K). The Mie channel is based on the fringe-imaging technique (McKay, 2002) where the Doppler frequency shift is measured from the spatial displacement of a linear interference pattern (fringe), produced by a FI, that is vertically imaged onto a detector. The latter is realized by an accumulation charge-coupled device (ACCD) with an array size of 16 pixels × 16 pixels (image zone). According to $\Delta f_{Doppler} = 2 f_0/c \cdot v_{LOS,}$ with $f_0 = 844.75$ THz being the laser emission frequency and $c$ being the speed of light, scattering from a particle that moves with a LOS (wind) speed of $v_{LOS,} = 1$ m·s$^{-1}$ introduces a Doppler frequency shift of about $\Delta f_{Doppler} = 5.63$ MHz. The shift manifests as a motion of the Mie fringe centroid position $m$ (Fig. 3 (b)) by about 0.056 pixels, given the nearly linear relationship between frequency and centroid position for typical wind speeds $v_{LOS}$ well below 100 m·s$^{-1}$ ($\Delta f_{Doppler} < 563$ MHz) and a slope of about 0.01 pixel·MHz$^{-1}$. The exact fringe position is calculated by a Nelder-Mead Downhill Simplex Algorithm (Nelder and Mead, 1965) that optimizes a Lorentzian line shape of the fringe intensity distribution after the electronic charges from all 16 rows in the image zone are binned together to one row with the integrated signal (see Fig. 3 (a)).

Measurement of the Doppler shift from the broadband molecular backscatter signal which features a Rayleigh-Brillouin line shape (Witschas, 2011a, b) is accomplished by the double-edge technique (Chanin et al., 1989; Garnier and Chanin, 1992; Flesia and Korb, 1999; Gentry et al., 2000). It relies on two bandpass filters (A and B) that are implemented as sequential Fabry-Pérot interferometers (FPIs) with adequate choice of spectral width and spacing with respect to each other. The signals transmitted through the two filters are imaged onto the Rayleigh channel detector as two nearly circular spots. After binning of the electronic charges from the 16 rows in the image zone, the charges from the left and right half of the resulting row are summed up. The contrast between the two total intensities is a measure of the spectral position of the backscatter spectrum with respect of the two fixed filter transmission curves, and hence allows determining the frequency shift between the emitted and backscattered laser pulse. Apart from the signal levels transmitted through the two filters, the respective centroid positions of the corresponding spots $r_A$ and $r_B$ (Fig. 3(c)) are relevant for the wind retrieval, as they provide information on the horizontal incidence angle of the backscattered light on the FPIs.

When the A2D is operating in so-called lidar mode, the summed-up signals from 25 images, i.e. 25 rows, are transferred to a memory zone of the ACCD one after another. Each row represents one range gate, from which three are used for detecting the solar background radiation (range gate #0), signals resulting from the voltage at the analogue-to-digital converter (detection chain offset, DCO, range gate #2) and the internal reference signal (range gate #4), respectively. Another three range gates act as buffers, so that 19 range gates are available for collecting the atmospheric backscatter signals (range gates #6 to #24). Owing to the transfer time from the image to the memory zone, the temporal resolution of one range gate is limited to 2.1 µs, which corresponds to a minimum height resolution of 296 m considering the 20° off-nadir viewing angle of the instrument. Moreover, it is important to note that the charge transfer process of the ACCD involves a temporal overlap in the acquisition of two subsequent range gates of about 1 µs. This issue is elaborated on in Sect. 3.3 in the context of the Mie wind errors and their reduction.





The ACCD allows to accumulate the signals from multiple successive laser pulses ($P$) on-chip to so-called "measurements", from which two pulses are lost due to the read-out of the ACCD. During the wind retrieval the signals from $N$ measurements are then summed up to one "observation" which always lasts 14 s, followed by 4 s for data read-out and transfer before the next observation starts. While a total of $P = 20$ pulses was emitted per measurement during the AVATARE campaign, corresponding to a measurement duration of 0.4 s, the number was decreased to $P = 10$ (0.2 s) for the AVATARI campaign.

Consequently, the number of measurements per observation was doubled from $N = 35$ to $N = 70$ which offered a higher flexibility of the quality control scheme that is applied on measurement level (see next section).

The performance of the A2D has been continuously improved over the past 16 years since it was put into operation in 2005. This encompasses first of all technical developments of the individual hardware components, particularly the UV laser. Here, the focus was laid on the improvement of the frequency and timing stability which was accomplished by a mechanical bridge

that connects the master oscillator cavity mirrors with a stiff construction of steel in order to enhance vibration tolerance as well as a novel active cavity control mechanism that represents a combination of the Ramp-Fire and the Ramp-Delay-Fire technique (Lemmerz et al., 2017). Thanks to these modifications, UV frequency stability of 1.8 MHz and timing stability below 100 ns were simultaneously achieved, thus forming the basis for highly accurate wind measurements. For comparison, the frequency stability of the ALADIN laser is on the order of 8 to 10 MHz (Lux et al., 2021). However, it was found that

other error sources limited the accuracy and precision of the A2D wind results. One major cause was identified to be speckle noise in the internal reference path which was introduced by the use of a fibre to transmit the internal reference signal from the laser to the front optics where it is injected into the receiver reception path and co-aligned with the atmospheric signal. Slow changes in the intensity distribution of the internal reference signal related to variations in laser frequency, polarization or (ambient) fibre temperature altered the Mie and Rayleigh internal reference response, and in turn, significantly increased

the random error. Consequently, a fibre scrambler was implemented into the A2D in preparation of the Cal/Val campaigns to overcome the detrimental speckle noise effect. As a result, the frequency variations of the internal reference signal were lowered by a factor of five (Mie) and two (Rayleigh), respectively, while intensity fluctuations were decreased by 55% (Mie) and 22% (Rayleigh) (Lux et al., 2019). In terms of the Rayleigh random error, which was assessed for the WindVal III campaign wind data with respect to the 2-µm DWL, a considerable reduction by 37% from 3.32 m·s$^{-1}$ to 2.08 m·s$^{-1}$ was

achieved (Lux et al., 2020a) thanks to the more stable internal reference.

Having overcome the limitation posed by the speckle noise, other deficiencies of the A2D Rayleigh and Mie channel that had been known before emerged more clearly. Regarding the Rayleigh channel, a major error source is related to the alignment sensitivity of the FPIs. Small variations of the incidence angle on the FPIs of a few percent of the total FOV, e.g. introduced by atmospheric inhomogeneity or beam pointing fluctuations, can lead to large wind errors of several m·s$^{-1}$ (DLR, 2012). This

effect is especially strong within the region of incomplete telescope overlap, i.e. in the near field of the instrument, as described above. For the Mie channel, the alignment sensitivity is much less critical, especially in the vertical axis of the fringe; however, a significant systematic error is introduced by the skewness of the imaged FI fringe, which stems from imperfections in the





optical system. Owing to the temporal overlap of the atmospheric range gates, strong backscatter gradients that fall within this range gate overlap can result in a shift of the fringe centroid position when the signal is summed up in the image zone. Consequently, the determined Mie responses of the two overlapping range gates are altered, thus deteriorating the Mie wind results.

The issues of the Rayleigh and Mie channel were tackled by refinements of the A2D wind retrieval which was advanced in the course of the AVATARE and AVATARI data analysis. While the Mie wind errors were reduced by a range gate averaging algorithm which corrects the Mie winds after the actual wind retrieval, the Rayleigh winds were improved by a novel quality control (QC) mechanism which was incorporated into the data processing. A more detailed explanation of the two aforementioned error sources and the counteracting retrieval modifications will be provided in two dedicated sections. Prior to that, the A2D wind processing is briefly described.

## 3.1 The A2D wind retrieval

A flow chart of the A2D processing chain used for the wind retrieval of a selected flight leg is shown in Fig. 4. After read-in of the required data (initialization file with processing settings, calibration data, digital elevation model (DEM) data, housekeeping data, detector signal data, aircraft data), geometrical calculations are performed to determine the range gate altitudes, LOS angles and velocity as well as the DEM intersection points. The latter are used within the ground detection algorithm. In a next step, the signal levels detected on the Rayleigh and Mie ACCDs are processed which includes the correction for the solar background and detection chain offset (DCO) as well as the consideration of the range and range gate thickness. Afterwards, QC procedures are applied to sort out invalid measurements, before the signals from all measurements within one observation are summed up. Up to this stage, the wind retrieval algorithm is identical for the Rayleigh and Mie channel. The transition from measurement level to observation level marks the point from which on different methods are applied to derive the wind speed according to the different measurement principles of the two channels.

Concerning the Mie channel, the Rayleigh background signal on the detector which is determined from the so-called MOUSR (Mie Out of Useful Spectral Range) procedure (Lux et al., 2020a) is subtracted prior to the determination of the Mie SNR (see definition in Marksteiner, 2013) for each bin of the matrix (*observation x range gate number*). Then, those bins for which the SNR is below the set threshold are flagged invalid (Mie mask). An independent Mie SNR threshold is set to create a second mask that distinguishes bins with high and low Mie SNR, i.e. high and low scattering ratio. This mask is inverted and applied on the Rayleigh channel to flag those bins invalid for which the Mie SNR exceeds a certain value (Rayleigh mask). Since the two Mie SNR thresholds can be set independently, a bin (and hence a wind result) can be valid in both the Rayleigh and Mie channel, namely when the Mie SNR lies between the two thresholds. This approach is similar to the grouping algorithm that is implemented in the Aeolus L2B processor for discriminating so-called Rayleigh-clear and Mie-cloudy winds (Tan et al., 2017; Rennie et al., 2020).





After the masking procedure, the ground detection is performed for both channels. The applied scheme is very similar to that
290   of Aeolus and was developed in the frame of the WindVal II data analysis (Weiler, 2017; Lux et al., 2018). The identification
of ground bins leads to additional masks for the two channels that further restrict the number of bins in the final wind curtain.
Next, the Mie and Rayleigh responses are derived for each bin through the determination of the Fizeau fringe centroid position
and the consideration of the FPI filter transmissions, respectively. Then, the Doppler frequency shift is calculated from the
atmospheric and internal reference path responses using the respective calibration data. Considering the LOS velocity of the
295   aircraft, one finally obtains the Mie and Rayleigh wind speed for each valid bin.

The modifications of the A2D wind retrieval that were implemented for mitigating and reducing the Rayleigh and Mie error
sources in this study are indicated as red-framed boxes in Fig. 4. They will be introduced in the following sections.

**Figure 4.** Flow chart of the A2D wind retrieval. Refinements that were introduced in the course of the AVATARE and AVATARI data
analysis are indicated as red-framed boxes. INT: internal path; ATM: atmospheric path; GR: ground return.



## 3.2 Mitigation of Rayleigh wind errors

As pointed out in previous publications on the A2D wind data retrieval (Lux et al., 2018, Lux et al., 2020a), one major aspect that limits the Rayleigh wind accuracy is the high sensitivity of the Rayleigh spectrometer response to variations in the incidence angle on the FPI. Despite the active transmit-receive co-alignment loop, atmospheric turbulence and the effect of strong cloud backscatter onto the co-alignment algorithm as well as laser beam pointing variations, either from inside the laser or from aircraft vibrations affecting the folding mirrors of the instrument, cause residual angular beam variations. Laboratory studies have shown that the Rayleigh response is primarily influenced by vertical incidence angle changes, whereas it is almost insensitive to angle changes along the horizontal axis of the FPI. In particular, it was demonstrated that vertical angular variations of the atmospheric beam of 1 µrad, i.e. 1% of the 100 µm FOV, causes an error in wind speed determination of about 0.4 m·s$^{-1}$ (DLR, 2012). The beam variations, as measured by a UV camera inside the A2D front optics (see Fig. 1 in Lux et al., 2018) are on the order of 5 to 10 µrad in both the horizontal and vertical direction in the case that the co-alignment loop is not disturbed, e.g. by cirrus clouds in the near field that cause saturation of the UV camera. Consequently, Rayleigh wind errors by at least 2 to 4 m·s$^{-1}$ are introduced by angular variations. The introduced error manifests as vertical stripes in the two-dimensional Rayleigh wind curtain, since it is correlated among the atmospheric range gates. The mean error varies from observation to observation according to the changes in the angle of incidence over several tens of seconds, which results in a fluctuating wind error from profile to profile.

This characteristic becomes also obvious from the example of the Aeolus underflight on 16 September 2019 in Fig. 5. The vertical pattern of the A2D Rayleigh curtain in panel (a) can be traced back to the angular variations. Comparison with the 2-µm DWL data, averaged onto the A2D measurement grid and projected onto the A2D LOS, in panel (b) allows deriving the wind error per bin which is shown in panel (c). The correlation of the A2D and 2-µm DWL wind data (panel (d)) reveals a positive bias which is mainly evident for winds that were measured at higher altitudes, i.e. in the near field of the instrument. The dependence of the Rayleigh wind error upon the altitude, as depicted in panel (e), illustrates this relation. The increasingly positive bias towards the instrument confirms the incomplete telescope overlap which reduces backscatter signal and emphasizes return from the large angle parts of the field, thus both aggravating the susceptibility of the Rayleigh channel to angular variations. Nevertheless, the increased sensitivity of the spectrometer response from near field backscatter signals to alignment changes can be exploited to correct for the associated wind error.



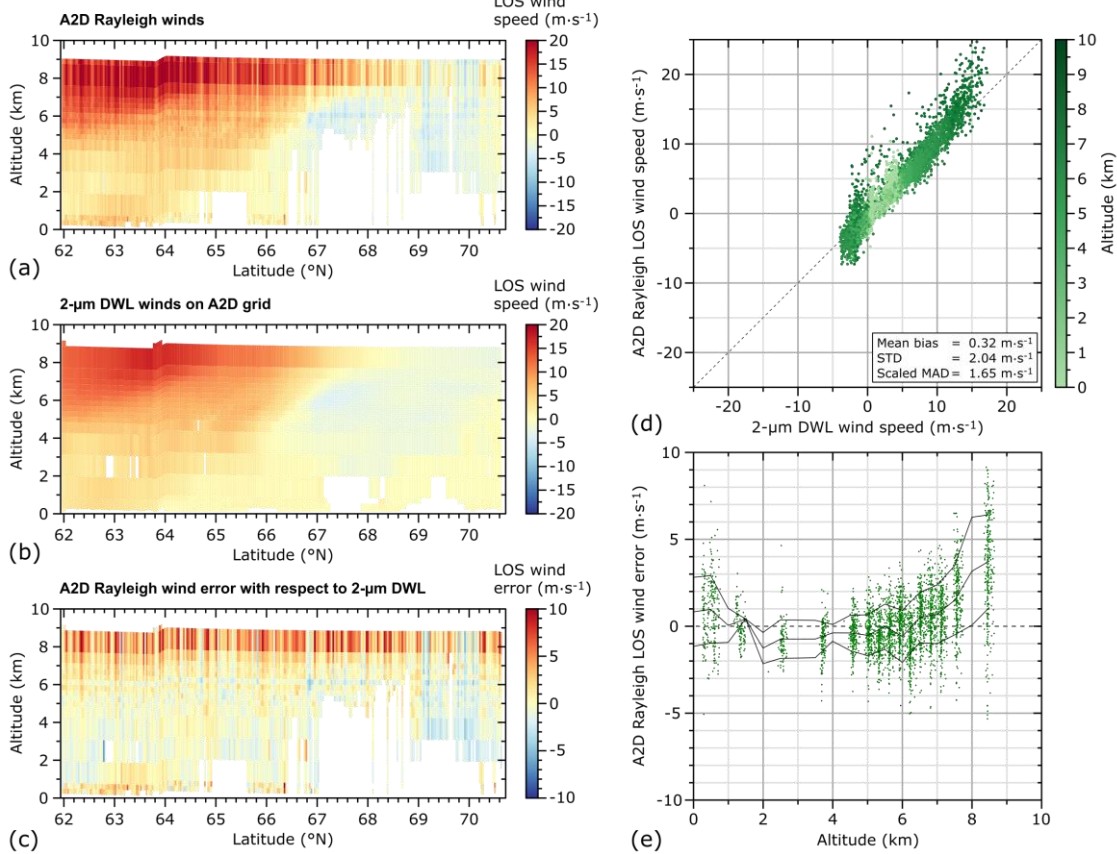

**Figure 5.** A2D Rayleigh wind for the AVATARI underflight on 16 September 2019 before the retrieval modifications: (a) A2D Rayleigh wind data, (b) 2-μm DWL wind data averaged onto the A2D measurement grid, (c) wind speed differences between (a) and (b), (d) scatterplots comparing the wind data from (a) and (b), where the color-coding describes the centre altitude of the considered range bin. Panel (e) depicts the altitude dependence of the A2D Rayleigh wind error with respect to the 2-μm DWL. The light green lines indicate the mean and standard deviation of the wind error in bins of 500 m.

The idea is to apply a QC mechanism which filters only those measurements within an observation for which the transmit path and receive path are well aligned and discard those measurements for which large deviations from the optimum alignment occur. This approach, however, requires a parameter that measures the quality of the co-alignment. In a first attempt to develop a QC scheme, the wind error with respect to the 2-μm DWL was correlated with the intensity of the backscatter signal. However, the correlation was found to be very poor so that the filtering of measurements according to the signal intensity did not improve the accuracy of the Rayleigh winds. Much better correlation was found between the wind error and the horizontal Rayleigh spot positions which are determined as the first moments of the intensities measured on pixels 0 to 7 (spot A, counted from 0) and pixels 8 to 15 (spot B) (Fig. 3 (c)). Interestingly, the signal intensities that are transmitted through the two FPI bandpass filters of the ALADIN Rayleigh channel during a dedicated calibration mode and their temporal evolution are also exploited for monitoring the Aeolus instrumental alignment and potentially ongoing drifts (Witschas et al., 2021).





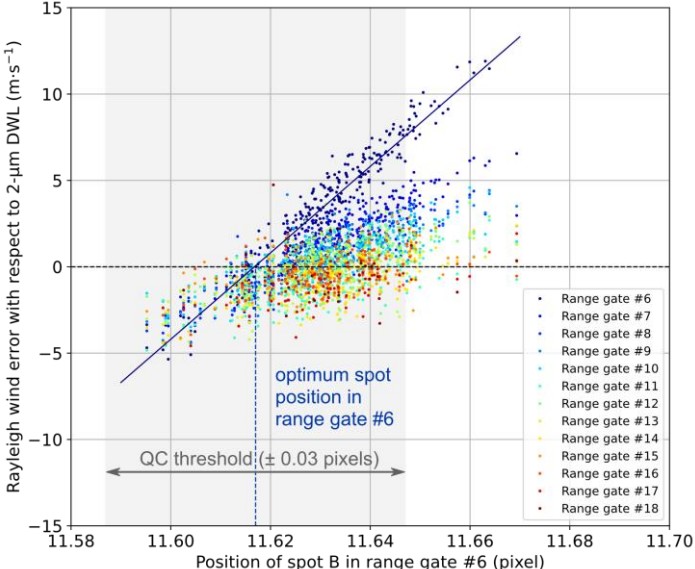

**Figure 6.** A2D Rayleigh wind error with respect to the 2-µm DWL in dependence on the horizontal position of spot B for the AVATARI underflight on 16 September 2019. The scatters are color-coded according to the A2D range gate. An optimum spot position is determined from those wind results in the near field (range gate #6, blue scatters) for which the wind error vanishes. Measurements for which the spot position in range gate #6 deviates from the optimum by more than 0.03 pixels are filtered out by the QC.

The scatterplot in Fig. 6 illustrates the good correlation of the wind error with the spot position, featuring almost linear relationships for different groups of scatters according to the range gate. Particularly in range gate #6, i.e. the second uppermost atmospheric range gate, a steep slope is apparent which confirms that the sensitivity of the Rayleigh spectrometer to angular variations is highest in the near field of the instrument. Note that the uppermost atmospheric range gate #5 is not useful due to the very small telescope overlap. For the purpose of filtering only measurements with good co-alignment, an optimum spot position is determined for which the wind error in the near field (range gate #6) vanishes. This procedure is illustrated in Fig. 6 for the underflight on 16 September 2019 where the optimum position of spot B is determined to be 11.617 pixels. Since the wind error is correlated among the range gates, the error also vanishes for those wind results that were measured in the other range gates at times when the spot position in range gate #6 is close to the determined optimum. Therefore, it is sufficient to only consider the most sensitive range gate #6 for the relationship between spot position and wind error. The reason why spot B is preferred over spot A is the higher sensitivity of its position to angular variations which was confirmed by analysing the correlation between spot position in range gate #6 and the A2D wind error for all Aeolus underflights from the AVATARE and AVATARI campaign. The scatterplots in Fig. 7 (a) and (b) show this relationship for the data from AVATARE and AVATARI, respectively, together with histograms that illustrate the distribution of spot positions for which the wind error is below ±0.5 m·s⁻¹. Note that the histograms include data from all atmospheric range gates in order to provide a sufficient amount of wind results for deriving meaningful statistics. As observed for the example in Fig. 6, a clear correlation between spot position and wind error is evident, which allows to precisely determine the optimum spot position for each flight leg. The latter are color-coded in the scatterplots, thus providing information on the variability of the optimum position during the campaigns.





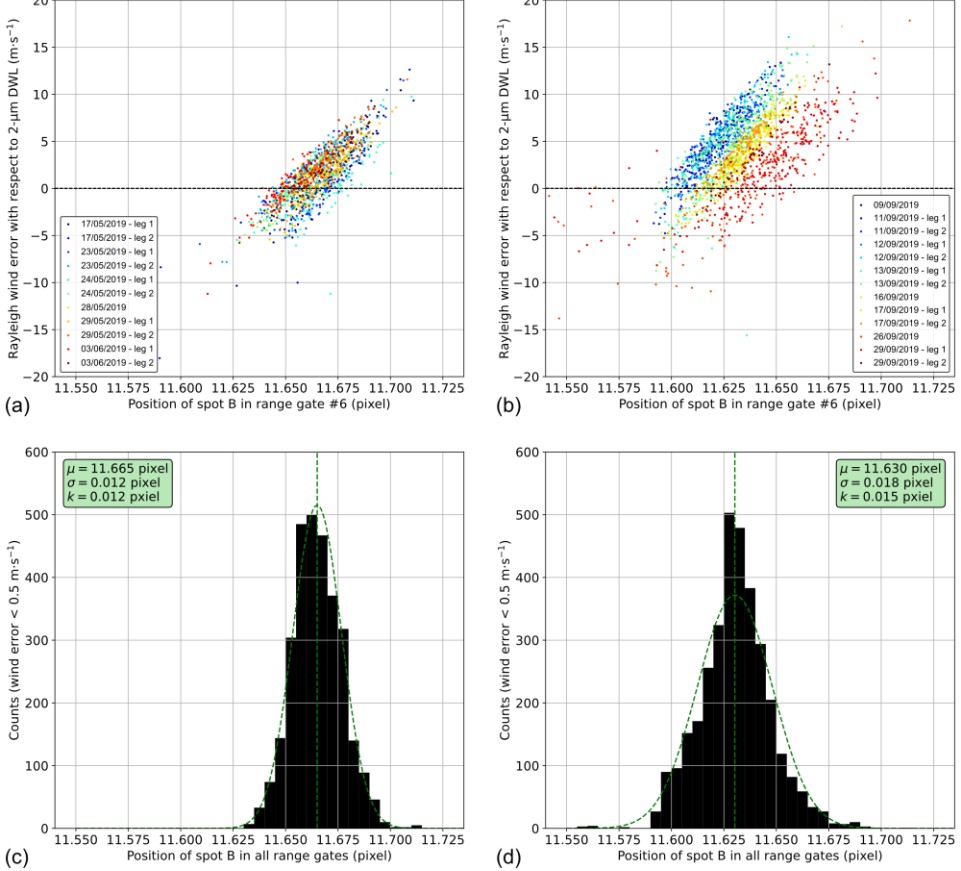

**Figure 7.** A2D Rayleigh wind error with respect to the 2-µm DWL in dependence on the position of spot B in range gate #6 for (a) the entire AVATARE campaign and (b) the entire AVATARI campaign (except for the underflights on 22/09/2019 and 23/09/2019). The scatters are color-coded according to the different Aeolus underflight legs. The bottom plots show histograms of spot positions from all range gates for which the wind error is $< \pm 0.5$ m·s$^{-1}$ (bin size: 0.005 pixel).

While the optimum position did not change much over the course of AVATARE, a clear trend to larger values is visible as the AVATARI campaign proceeded. This is most likely related to an alignment drift that occurred over the nearly three weeks between the first and last underflight and also manifests in a wider distribution of "ideal" spot positions in the histogram (Fig. 7 (d)). It is important to note that, in addition to the horizontal spot motions which are analysed here, alignment changes can also involve a vertical motion of the Rayleigh spots. Such motions can, however, not be detected after summation of the signals from the image zone to one line in lidar mode, as explained in Sect. 3. Sensitivity studies have revealed that variations in the vertical spot position cause much stronger systematic error than horizontal motions. Comparison of the two campaign datasets also shows that the respective optimum spot positions slightly deviate from each other (AVATARE: 11.67 pixels, AVATARI: 11.63 pixels), suggesting a very small change in the instrument alignment. However, considering that the instrument was dismounted and reintegrated into the Falcon aircraft in between, the reproducibility of the optimum is rather impressive.





For the actual QC procedure, the optimum spot position is determined for each flight leg, i.e. wind scene, individually. Then, measurements for which the spot position deviates by more than 0.03 pixels are rejected before the summation of signals to

the observational level. The threshold value of 0.03 pixels was found to provide the best filtering in terms of the improvement of the wind results. At larger thresholds, the QC is less efficient; at smaller thresholds, the number of measurements that pass the QC is so small that the signal levels per observation become too low and the wind results are significantly affected by noise. This corresponds to 2% allowed alignment fluctuations of the 1.5 pixels spot width, which corresponds to a 100 µrad FOV in the atmosphere. Hence, only measurements for which the horizontal beam variations are less than about ±2 µrad from

the ideal incidence angle on the FPI pass the QC. The UV camera data used for the transmit-receive co-alignment loop shows that the angular variations have comparable amplitude in both beam directions for most of the underflights. Consequently, the applied QC approach, in principle, limits the Rayleigh wind error introduced by (vertical) angular variations to about 4 µrad · 0.4 m·s$^{-1}$/µrad = 1.6 m·s$^{-1}$.

Regarding all wind scenes from the two campaigns, the portion of invalid measurements is typically between 30% and 60%,

underlining the significant impact of the QC on the data processing. It should be noted that the altitude-dependent relationship between the wind error and the spot position in range gate #6 is not always as linear as for the example shown in Fig. 6. Moreover, in some cases, only very few wind results are available in some of the range gates beyond #6, resulting in a poor correlation. Therefore, the determination of altitude-dependent linear regression functions for performing a wind correction instead of a QC scheme is difficult to be implemented in the wind retrieval. Note also that the described QC is likewise applied

to Mie measurements although the Mie response is less sensitive to alignment variations. Nevertheless, it is considered reasonable to remove measurements with suboptimal co-alignment from the Mie data as well, although the impact is found to be very small.

The effect of the new QC on the wind results from 16 September 2019 is depicted in Fig. 8. The Rayleigh curtain is apparently much smoother than without the QC, as the latter significantly diminishes the wind error, and hence the magnitude of the error

variations from observation to observation that produce the vertical stripes of the curtain. The reduction in wind error is also clearly visible in the scatterplot which shows a much better correlation between the A2D Rayleigh and the 2-µm DWL reference winds, although the mean bias is slightly increased from 0.32 m·s$^{-1}$ to -0.47 m·s$^{-1}$. However, the standard deviation and the scaled MAD are decreased by 25% from 2.04 m·s$^{-1}$ to 1.54 m·s$^{-1}$ and by 17% from 1.65 m·s$^{-1}$ to 1.37 m·s$^{-1}$, respectively. The improvement is most pronounced in the near field winds at altitudes above 7 km where the large positive bias is effectively

reduced by several m·s$^{-1}$, albeit a small residual trend to positive biases is still evident towards higher altitudes on panel (e) of Fig. 8. For instance, the bias of those winds that were measured in the uppermost atmospheric range gate at altitudes around 8.5 km is decreased from 3.71 m·s$^{-1}$ to 1.49 m·s$^{-1}$. Consequently, the new QC allows exploiting the full vertical measurement range of the A2D, whereas the wind data from the first 1.5 to 2 km below the aircraft had to be discarded in previous campaigns due to the large systematic error.



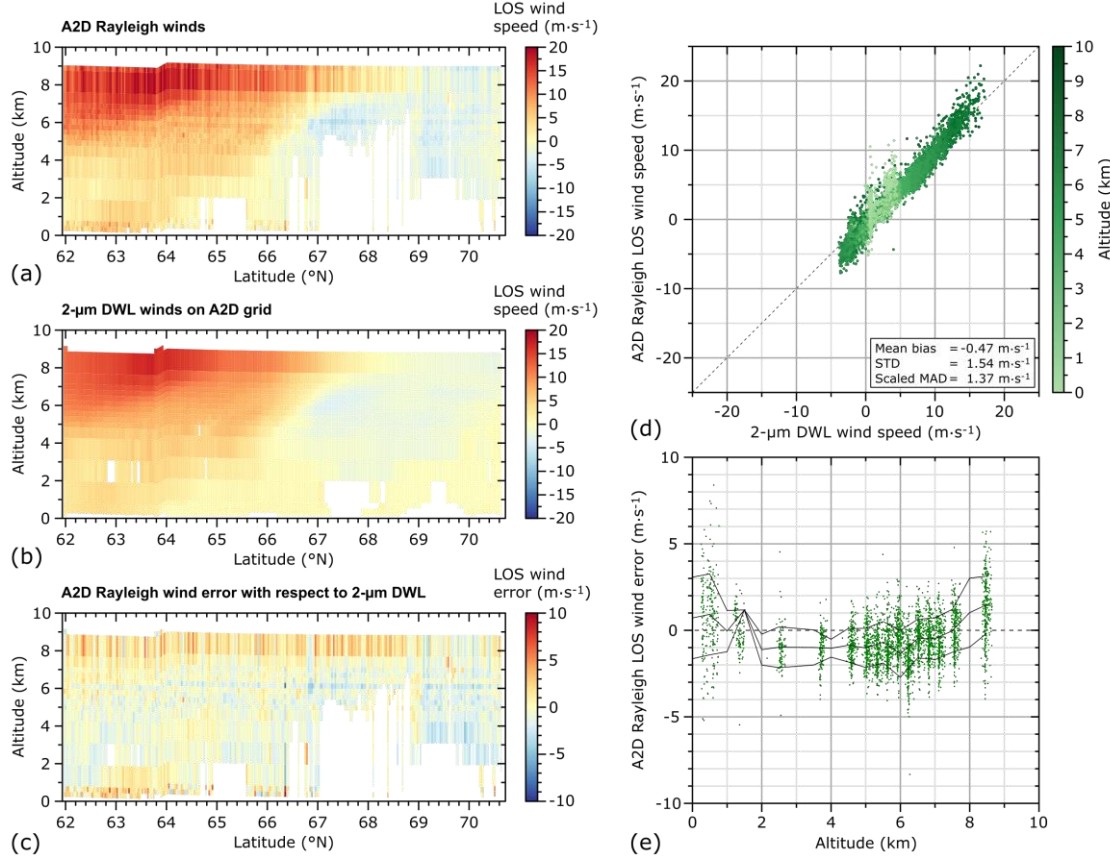

**Figure 8.** A2D Rayleigh wind for the underflight on 16 September 2019 as in Fig. 5 but with usage of the novel spot position QC.

The developed QC mechanism requires the 2-µm DWL wind data as a reference to determine the optimum spot position for each flight leg. However, it should be mentioned that the optimum values vary by only about 0.02 pixels from leg to leg, except for the flights on 22/09 and 23/09 where the position of spot B were around 12.1 pixels. This is probably due to changes in the alignment that had occurred after a cooling of the instrument in the hangar, and the data from these two flights was therefore excluded from the statistics. Hence, the utilization of a fixed optimum for an entire campaign still leads to an improvement of the wind quality. The impact was found to be about half as large in this case.

### 3.3 Reduction of Mie wind errors

The major error source which currently limits the accuracy of the A2D Mie wind results is of different origin than that of the Rayleigh channel. The Mie response, i.e. the fringe centroid position, is much less sensitive to variations of the incidence angle on the FI. However, it is affected by the shape of the fringe in combination with the temporal range gate overlap of the ACCD detector which has been introduced above. Imperfections of the FI and its illumination cause aberrations such that the fringe is not a straight line, but slightly skewed and/or not evenly illuminated from top to bottom. This effect is more pronounced for



the A2D compared to ALADIN where the fringe exhibits a smaller degree of skewness. The implications of the Mie skewness

for the A2D Mie wind bias is illustrated in Fig. 9. The Mie fringe is imaged onto the 16 x 16 pixel image zone of the ACCD which is continuously illuminated by the atmospheric backscatter signals. One image corresponds to a predefined range gate, where the vertical resolution of the range gates is determined by the sampling time for one image. For instance, an integration time of $\Delta t = 2.1$ µs corresponds to a range resolution of $c \cdot \Delta t/2 = 315$ m or height resolution of 315 m $\cdot \cos(20°) = 296$ m, respectively. The signal of one range gate is summed up by pushing down the charges row by row into the transfer row of the

ACCD, while the signal from the subjacent range gate already appears on the image zone. This process takes 1 µs which leads to a range gate overlap and a vertical smearing of the signal. Consequently, the signal summation in the transfer row is influenced by the signal intensity of each line in the image zone during the transfer process, and hence by the scattering ratio in the range gate overlap region.

In case of equally strong or weak backscatter in the overlapping range gates (left column of Fig. 9), the fringe centroid position

$m$, which is determined from the summed signal intensities on the 16 transfer row pixels, is not affected. The same holds true for the situation when strong backscatter signals, e.g. from a thick cloud, are present in only one range gate (middle column). However, when a thick cloud falls within the range gate overlap region (right column), the contribution of the strong backscatter signals is divided among the neighbouring range gates. The lower part of the fringe image for range gate #$N$ is more intense than the upper part, whereas it is vice-versa for range gate #($N$+1). Owing to the fringe skewness, the different

weighting of the 16 lines of the fringe alters the summed signal distribution in the transfer row. As a result, the derived centroid positions $m'$ and $m''$ are down- and upshifted in frequency, respectively, leading to opposing biases of the wind speeds that are then retrieved for the two adjacent range gates. Note that this effect on the Mie responses does not occur for a straight fringe, as a potential different weighting of the 16 lines of the image zone would not affect the signal distribution in the transfer row. Interestingly, a similar effect is observed for coherent DWLs and radar instruments at cloud boundaries in conjunction

with laser chirp causing artefacts in the measured wind data and introducing signal offsets (Bühl et al., 2012).

The described effect not only occurs in instances when strong backscatter gradients fall into the range gate overlap region, but also in case of strong vertical wind gradients that cannot be resolved by the A2D. This also results in a "dipole" structure of the wind bias, as the fringe position is under- and overestimated in neighbouring range gates, respectively, even in absence of a fringe skewness. Therefore, systematic errors introduced by the range gate overlap in combination with strong vertical wind

gradients are likely to be also present for ALADIN.





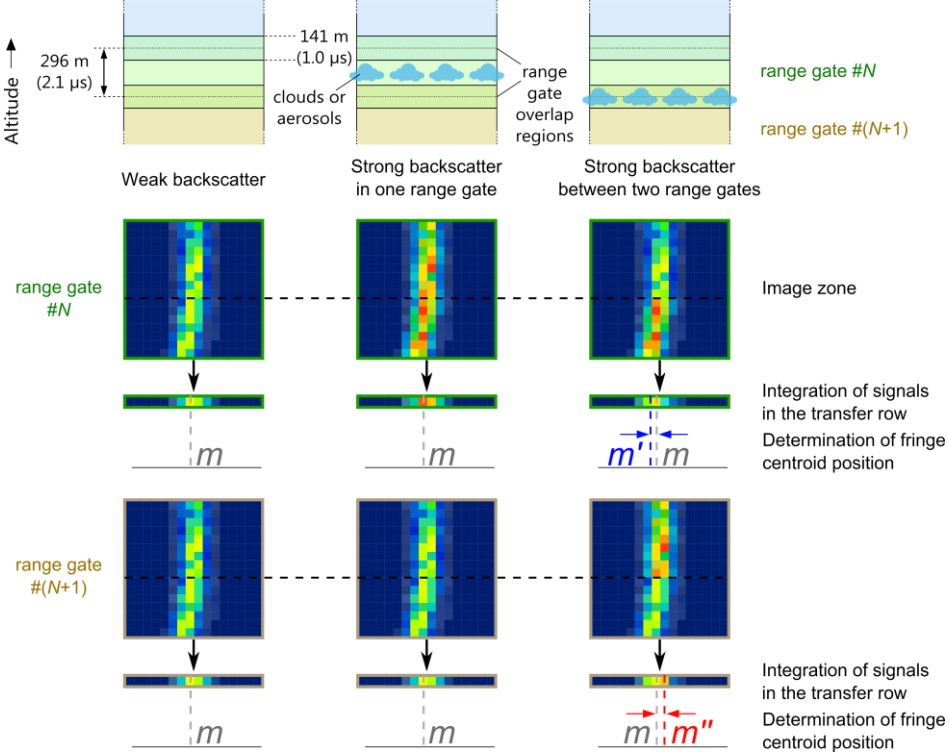

**Figure 9.** Illustration of Mie wind errors due to fringe skewness (see text for explanation).

The consequence of the fringe skewness on the Mie wind results is presented at the example of the first underflight leg on 23 May 2019 in Fig. 10. During this wind scene, dense, patchy clouds were encountered at different altitudes, resulting in strong backscatter signals distributed over multiple range gates. This caused large positive and negative wind biases at the cloud edges, i.e. where strong backscatter gradients occurred. The opposing biases are clearly visible as red and blue bins in the Mie wind curtain (panel (a)) as well as in the corresponding scatterplot comparing the wind results with the 2-μm DWL reference wind data (panel (c)).

In order to correct for the large wind errors, the affected bins were first identified by determining the vertical wind speed gradient. The corresponding curtain is shown in panel (b) of Fig. 10. In a next step, bins for which the absolute gradient is above 5 m·s$^{-1}$·km$^{-1}$ (LOS), the wind speeds from the flagged bin and the one above are averaged. Since real LOS wind gradients are typically below 5 m·s$^{-1}$·km$^{-1}$, even in the area of jet streams, the chosen threshold was found to reliably identify the erroneous bins while unaffected bins remained unflagged. The averaged wind speed is then allocated to the two contributing bins. In this way, the large positive and negative biases (partly) compensate each other. As a result, the spread in the wind speeds of adjacent erroneous bins is reduced, leading to higher precision of the Mie winds. For the selected wind scene, the standard deviation is decreased by nearly 40% from 3.82 m·s$^{-1}$ to 2.33 m·s$^{-1}$. The impact of the bin averaging method on the Mie data quality of the entire AVATARE and AVATARI campaigns is discussed in Sect. 4.

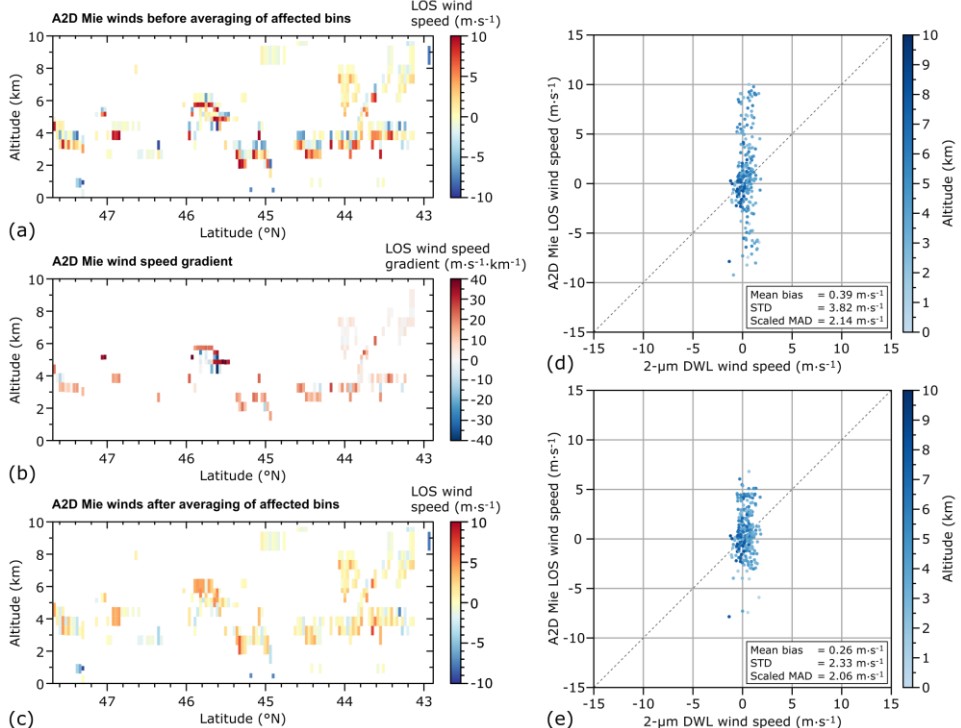

**Figure 10.** Mie winds for the AVATARE underflight on 23 May 2019 before and after retrieval modifications: (a) A2D Mie winds without bin averaging, (b) Mie wind gradient without bin averaging, (c) Mie winds with bin averaging. The right scatterplots show the correlation between the Mie winds and the 2-µm DWL wind data without (d) and with (e) bin averaging.

The presented technique can be regarded as a post-processing of the Mie wind results, as it is applied after the actual wind retrieval. Moreover, this technique is only applicable when at least two valid adjacent vertical Mie bins are available. In case the Mie SNR is too low in one of the overlapping range bins and it is thus flagged invalid, no information on the wind speed gradient is available and the other bin suffering the fringe skewness error cannot be corrected. Such cases of isolated erroneous bins are visible in the wind curtains in Fig. 10 at about 1 km altitude. Hence, as an alternative approach, it was also tested to identify the affected bins by checking the Mie SNR gradient which is available for all bins. However, the correlation of the Mie SNR gradient with the wind error was found to be subtle and complicated case-by-case analysis was necessary to sort out the largest wind speed outliers while retaining the wind results which agreed well with the 2-µm DWL wind data. This fact underlines that the error introduced by the Mie fringe skewness in combination with the range gate overlap is very complex. Therefore, it was decided to use the simple bin averaging approach to improve the Mie wind data quality.

It is interesting to note that preliminary analysis of selected Aeolus Mie wind profiles also revealed opposing wind biases in vertically neighbouring range bins, particularly in lower altitudes. This suggests that, despite the smaller fringe skewness observed for the satellite instrument, a similar error source is present for Aeolus. However, a more detailed investigation of larger Aeolus datasets is required to verify this assumption. Here, it is important to differentiate between wind scenes with strong vertical wind speed gradients, those with strong vertical backscatter gradients as well as those for which both occur.



### 3.4 A2D error assessment using the 2-µm DWL

The accuracy and precision of the A2D Mie and Rayleigh wind results from the two airborne campaigns were evaluated by
comparing them to the wind data obtained with the coherent 2-µm DWL. The latter offers high accuracy of the horizontal wind
speed of about 0.1 m·s⁻¹ and precision of better than 1 m·s⁻¹ (Weissmann et al. 2005; Witschas et al., 2017) so that it is
considered as the true reference. For adequate comparison between the two different lidar datasets, the three-dimensional wind
vectors measured with the 2-µm DWL were projected onto the A2D LOS axis. Moreover, the 2-µm measurement grid was
adapted to that of the A2D by means of the weighted aerial interpolation algorithm that was also applied for the analysis of the
A2D wind data quality of previous campaigns (Marksteiner, 2013; Lux et al., 2018; Lux et al., 2020a). After harmonization of
the two datasets, a bin-to-bin comparison was performed to derive the systematic and random error of the A2D winds for all
underflights of the respective campaign, whilst assuming that the representativeness error is negligible. Note that the wind
results from additional flight legs aside from the Aeolus measurement swath did not enter the statistical comparison. The error
assessment was performed separately for the AVATARE and AVATARI campaign, as well as before and after implementation
of the retrieval modifications discussed above to demonstrate their impact on the A2D wind data quality.

### 3.4.1 AVATARE

The results from the statistical comparison of the A2D Mie and Rayleigh LOS winds with the 2-µm DWL are presented as
probability density functions (PDFs) of the wind speed differences in Fig. 11. Panel (a) shows that, for the AVATARE dataset,
the Mie wind errors that are related to the fringe skewness are largely diminished by averaging the winds in adjacent range
gates with extraordinarily high (opposing) bias, as described in Sect. 3.3. Thanks to this method the random error, represented
by the scaled median absolute deviation (MAD), is reduced from 1.61 m·s⁻¹ to 1.47 m·s⁻¹, while the mean bias is also decreased
from -0.42 m·s⁻¹ to -0.32 m·s⁻¹. The effect on the standard deviation is even larger (-22% from 2.23 m·s⁻¹ to 1.73 m·s⁻¹), as this
value is more sensitive to the outliers that are caused by the fringe skewness issue than the scaled MAD. Another parameter
that illustrates the positive effect of the retrieval changes is the number of gross errors which are defined as those wind results
that deviate by more than four times the scaled MAD from the 2-µm DWL wind speed. Note that the scaled MAD values
shown in the insets in Fig. 11 are calculated after discarding the gross errors which are indicated as red bars in the histograms.
Hence, the original scaled MAD values considering all A2D wind results are usually larger. For instance, the complete set of
measured Mie winds before processing modifications shows a scaled MAD of 1.75 m·s⁻¹. Consequently, wind results that
differ by more than 4 · 1.75 m·s⁻¹ = 7.00 m·s⁻¹ from the respective 2-µm DWL wind speed in the compared bin are considered
as gross errors and excluded from the statistical calculations, resulting in a new scaled MAD value of 1.61 m·s⁻¹. The number
of Mie gross wind errors acquired during the AVATARE campaign when using the original A2D wind retrieval is 227, and is
drastically reduced to 42 upon implementing the bin averaging method. At the same time, the PDF is transformed to a much
more Gaussian contribution which clearly demonstrates the improvement of the A2D wind accuracy.





**Figure 11.** Probability density functions for the A2D Mie (a) and Rayleigh (b) wind error with respect to the 2-µm DWL for the entire AVATARE campaign before (top plot in each panel) and after (bottom plot) the applied retrieval modifications. Gross errors (see definition in the text) are marked in red. The dashed lines represent Gaussian fits. The insets summarize the mean values µ, standard deviations σ and scaled MAD $k$ calculated for the respective datasets. The corresponding PDFs of the A2D Mie and Rayleigh wind error for the AVATARI dataset are provided in panels (c) and (d), respectively.



The impact of the retrieval modifications on the Rayleigh wind results is rather small for the AVATARE dataset (panel (b)). The mean bias is changed from 0.21 m·s⁻¹ to -0.14 m·s⁻¹, as a significant amount of the positively biased winds measured at higher altitudes (see Fig. 5 (e)) is corrected by the spot position QC. The latter also reduces the random error from 1.44 m·s⁻¹ to 1.28 m·s⁻¹, while the number of gross errors is slightly decreased from 134 to 130.

An interesting aspect of the A2D wind quality assessment is the altitude dependence of the wind error. Here, the analysis

showed that the reduction of the systematic and random Mie wind errors by means of the retrieval changes primarily acts on the winds measured below 6 km (see also Fig. 10). This can be explained by the fact that thicker clouds that give rise to the error source related to the fringe skewness are more present in the lower and middle troposphere. The situation is opposite for the Rayleigh channel, where mainly those winds that were measured at higher altitudes above 7 km are improved in quality by the made modifications. This is due to the nature of the developed QC based on the Rayleigh spot positions which

predominantly tackles the error introduced by the incomplete telescope overlap, i.e. in the near field of the instrument. It was also found that the Mie random error is smallest at altitudes above 6 km, probably because of the presence of thin cirrus clouds. The latter are favourable for accurate Mie wind retrieval, as they provide high Mie SNR without sharp boundaries which exacerbate the Mie fringe skewness effect. On the other hand, cirrus clouds are detrimental to the transmit-receive co-alignment loop, as they affect the determination of the return spot position on the UV camera in the front optics, thereby introducing

higher angular variations of the beam being incident on the Rayleigh spectrometers and, in turn, larger Rayleigh random errors. This effect contributes to the slightly increased random error at high altitudes.

### 3.4.2 AVATARI

Given the fact that almost twice as many Aeolus underflights were performed in the frame of the AVATARI campaign, the latter provided an even larger A2D dataset compared to the AVATARE campaign. Aside from the higher number of winds,

the range of detected LOS wind speeds was considerably higher during AVATARI (≈ ±20 m·s⁻¹) than those observed during AVATARE (≈ ±15 m·s⁻¹), since many of the AVATARI underflights targeted the sampling of the North Atlantic jet stream. This allowed assessing the A2D wind data quality over a larger wind speed range. The PDFs derived from the statistical comparisons of the A2D Mie and Rayleigh LOS winds with the 2-µm DWL data before and after the retrieval refinements are depicted in Fig. 11 (c) and (d). Concerning the Mie channel (panel (c)), the systematic error is reduced from -1.07 m·s⁻¹ to -

0.15 m·s⁻¹, which is however not only due to the modifications. In addition, a so-called Zero Wind Correction (ZWC) was performed which involves the analysis of strong ground return signals over ice that were collected during two underflights along the east coast of Greenland. Here, the measured Mie ground velocity was found to be (-0.9 ± 0.5) m·s⁻¹ which pointed to a systematic error source of the A2D Mie channel during the AVATARI campaign. This assumption was further strengthened by the fact that the A2D Mie winds were negatively biased by about 1 m·s⁻¹ with respect to the 2-µm DWL for

most of the underflights. Hence, the determined ZWC value of -0.9 m·s⁻¹ was subtracted from all measured Mie wind speeds of the AVATARI dataset. A likewise verification or correction of the systematic Rayleigh wind error was inhibited by the



much higher noise of the Rayleigh ground velocities. The same is true for both the Rayleigh and Mie wind data from the AVATARE campaign, as there were no flights over high-albedo surfaces and thus no sufficiently strong ground return for applying a ZWC. While the ZWC reduced the Mie systematic error, the Mie random error is decreased by the mitigation of

the fringe skewness error from 1.35 m·s⁻¹ to 1.27 m·s⁻¹. Like for the AVATARE campaign, this correction mainly affected those winds that were measured at lower altitudes, i.e. in regions where thicker clouds resided that are likely to cause the fringe skewness error. For the Rayleigh winds, a large improvement is apparent both in terms of accuracy and precision. This can primarily be traced back to the correction of the positively biased higher altitude (i.e. near field) winds by the spot position QC which diminishes the mean bias from 0.90 m·s⁻¹ to 0.13 m·s⁻¹ while the random error decreased from 2.12 m·s⁻¹ to 1.78 m·s⁻¹.

When only considering winds measured at altitudes above 8 km, the mean bias is decreased by nearly 70% from 2.08 m·s⁻¹ to 0.65 m·s⁻¹. The considerable improvement of the A2D wind accuracy and precision also manifests significant reduction of gross errors (Mie: -52%, Rayleigh: -42%). Finally, the distributions of the Mie and Rayleigh wind errors with respect to the 2-μm DWL become more Gaussian upon the retrieval modifications, thus verifying its positive impact on the wind data quality.

### 3.4.3 Comparison with previous campaigns

The large impact of the retrieval modifications on the A2D wind data accuracy and precision which was pointed out in the previous sections also becomes obvious when looking back at the error assessments for the previous airborne campaigns. For this purpose, the results from the statistical comparisons with the 2-μm DWL for AVATARE and AVATARI are summarized in Table 3 (Mie) and Table 4 (Rayleigh) together with the corresponding results for the WindVal II (2016) and WindVal III (2018). When comparing the number of gross errors, it should be noted that for WindVal II and WindVal III, gross errors were

defined as winds that deviated by more than 10 m·s⁻¹ from the 2-μm DWL winds, whereas for the recent campaigns they were defined as winds that deviate by more than four times the scaled MAD which is always below 10 m·s⁻¹. Most notably, the A2D random errors, which typically ranged between 2 and 3 m·s⁻¹ for both channels depending on the encountered atmospheric conditions during the previous campaigns, were drastically reduced to values of (1.5 ± 0.3) m·s⁻¹. Moreover, it should be emphasized that the A2D Rayleigh wind data entering the statistical comparisons for the previous campaigns was restricted to

altitudes below 7.5 km due to the known large systematic error in the near field which has now been overcome thanks to the spot position QC. In this respect, the retrieval changes have not only improved the accuracy and precision of the Rayleigh channel, but also markedly increased its effective measurement range to higher altitudes. This is especially important for the Aeolus validation, as the range between 7.5 km and the flight altitude (10 to 12 km) is the region where strong wind gradients and high wind speeds, e.g. related to the jet stream, typically occur. Such features are well-suited for evaluating the data quality

of the Aeolus wind product. Regarding the Mie channel, the made modifications primarily improved the quality of wind data that was acquired in scenes where thick clouds promoted the fringe skewness error, resulting in a large reduction of the random error compared to previous campaigns. In summary, the accuracy and precision of the A2D wind data form an excellent basis for the Aeolus validation which will be presented in the next section.



**Table 3.** Results of the statistical comparison between the A2D Mie and the 2-µm LOS wind data obtained from the WindVal II, WindVal III (without retrieval improvements) as well as the AVATARE and AVATARI campaigns (with retrieval improvements). For WindVal II and WindVal III, gross errors are defined as winds that deviate by more than 10 m·s⁻¹ from the 2-µm winds. For AVATARE and AVATARI gross errors are defined as winds that deviate by more than 4 times the scaled MAD from the 2-µm winds.

| Statistical parameter | WindVal II | WindVal III | AVATARE | AVATARI |
|---|---|---|---|---|
| Number of compared bins | 7169 | 951 | 3735 | 7486 |
| Number of gross errors | 5 (0.07%) | 9 (0.52%) | 42 (0.13%) | 119 (0.07%) |
| Correlation coefficient $r$ | 0.98 | 0.68 | 0.94 | 0.97 |
| Mean bias (A2D – 2-µm) | 0.53 m·s⁻¹ | 1.18 m·s⁻¹ | -0.32 m·s⁻¹ | -0.15 m·s⁻¹ |
| Standard deviation | 2.03 m·s⁻¹ | 2.91 m·s⁻¹ | 1.73 m·s⁻¹ | 1.50 m·s⁻¹ |
| Scaled MAD | 1.79 m·s⁻¹ | 2.37 m·s⁻¹ | 1.47 m·s⁻¹ | 1.27 m·s⁻¹ |

**Table 4.** As Table 3 but for the statistical comparison between the A2D Rayleigh and the 2-µm LOS wind data.

| Statistical parameter | WindVal II (see Lux et al., 2020a) | WindVal III (see Lux et al., 2020a) | AVATARE | AVATARI |
|---|---|---|---|---|
| Number of compared bins | 2575 | 1301 | 14396 | 22760 |
| Number of gross errors | 308 (0.17%) | 36 (0.37%) | 130 (0.03%) | 163 (0.02%) |
| Correlation coefficient $r$ | 0.94 | 0.83 | 0.95 | 0.96 |
| Mean bias (A2D – 2-µm) | -0.47 m·s⁻¹ | -0.35 m·s⁻¹ | -0.14 m·s⁻¹ | 0.13 m·s⁻¹ |
| Standard deviation | 3.32 m·s⁻¹ | 2.08 m·s⁻¹ | 1.45 m·s⁻¹ | 1.90 m·s⁻¹ |
| Scaled MAD | 2.91 m·s⁻¹ | 1.94 m·s⁻¹ | 1.28 m·s⁻¹ | 1.78 m·s⁻¹ |

## 4 Aeolus validation using the improved A2D

From the WindVal III campaign only a small dataset was available for the Aeolus validation, as the A2D was operable during only three underflights. Moreover, the range gate settings of both ALADIN and the A2D were not optimal with many narrow range gates in the lower troposphere, where Aeolus winds often exhibited large wind errors and were therefore discarded from the analysis (Lux et al., 2020a), resulting in a rather small number of validated Aeolus wind results. The latter were also only of preliminary nature with relatively large systematic and random errors early in the mission. In this sense, the Aeolus error assessment that is presented in the following section provides a much higher validity than that of the previous work.

The A2D Mie and Rayleigh wind data collected along the Aeolus measurement swaths during the 16 AVATARE and AVATARI underflights was shown to be of good quality with mean biases of less than 0.2 m·s⁻¹ (except for the Mie winds of the AVATARE campaign with -0.32 m·s⁻¹) and random errors of less than 1.8 m·s⁻¹ with respect to the 2-µm DWL. These values represent the real instrument error of the A2D which measures LOS wind speeds at an off-nadir angle of 20°. The Aeolus wind product, however, contains the horizontal LOS (HLOS) wind speeds. Hence, it was decided to convert the A2D LOS wind speeds to HLOS wind speeds for the purpose of comparing the two datasets with each other. Adequate comparison



also requires to consider the different viewing geometries (or pointing angles) of the satellite instrument and its airborne demonstrator. Furthermore, the different horizontal and vertical resolutions of the two instruments necessitate an adaptation

of the A2D measurement grid to that of Aeolus. The procedures to harmonize the two datasets are comprehensively described in Lux et al. (2020a). The conversion of the A2D LOS winds to the satellite HLOS involved the use of wind data from the 2-µm DWL (zonal and meridional wind components) and the ECMWF model background that is included in the Aeolus AUX_MET data product which contains wind vector information along the predicted Aeolus track (Rennie et al., 2020). The latter was only used in those parts of the A2D data coverage where valid 2-µm DWL winds were not available, particularly in

atmospheric regions with very weak particle backscatter signals.

The quality of the Aeolus wind data contained in the L2B product was considerably improved by the correction for large systematic errors which had strongly degraded the wind data quality in the initial phase of the mission (Kanitz et al., 2020; Reitebuch et al., 2020). While dark current signal anomalies on single ("hot") pixels of the Aeolus detectors were successfully mitigated for by the implementation of dedicated calibration instrument modes (Weiler et al., 2020), wind biases that were

introduced by variations in the temperature distribution across the primary telescope mirror were largely diminished by the so-called M1 correction based on ECMWF model winds (Rennie and Isaksen, 2020; Weiler et al., 2021; Rennie et al., 2021). These corrections formed the basis for the operational assimilation of the Aeolus wind data in NWP by various weather services and were also applied for the reprocessing (baseline 10) of the Aeolus wind products that cover the period of the AVATARI campaign in September 2019 (Abdalla et al., 2020; Masoumzadeh et al., 2020). However, reprocessed wind data for the

April/May 2019 time frame when the AVATARE campaign was conducted will only become available in 2022. Therefore, the statistical comparison of the Aeolus wind data with the A2D winds in this section is restricted to the AVATARI dataset.

In contrast to the AVATARE and WindVal III validation campaigns (Lux et al., 2020a), satellite underflights were not only carried out both on ascending orbits of Aeolus, but also along descending orbits. Since the systematic and random errors were found to deviate for the two satellite directions, wind data from the four flights along descending orbits (16/09/2019,

17/09/2019, 22/09/2019 and 26/09/2019, see also Table 2) and the six flights along ascending orbits were analysed separately. The scatterplots comparing the Mie and Rayleigh wind data from the two direct-detection wind lidars for ascending orbits (top row) and descending orbits of Aeolus (bottom row) are presented in Fig. 12.

Several filters were applied to the two datasets. While gross errors, as defined in section 3.4.1, were removed from the A2D dataset, Aeolus winds with an estimated HLOS wind error larger than 8 m·s$^{-1}$ (Rayleigh-clear) and larger than 4 m·s$^{-1}$ (Mie-

cloudy) were discarded. These thresholds were found to ensure small departures of the Aeolus winds from the high-accuracy 2-µm DWL wind data (Witschas et al., 2020). The estimated wind error is included in the L2B product and is primarily related to the SNR for the Mie channel, while the pressure and temperature sensitivity of the Rayleigh responses is considered for the Rayleigh channel in addition to the SNR (Rennie et al., 2020). The latter depends on the return signal level and the solar background level. Finally, only Aeolus winds in bins which are covered by at least 25% with A2D bins entered the statistical

comparison. Optimization of this so-called coverage ratio was discussed in Lux et al. (2020a) in the context of the WindVal III campaign results and ensures a sufficient number of compared bins while preventing large representativity errors.





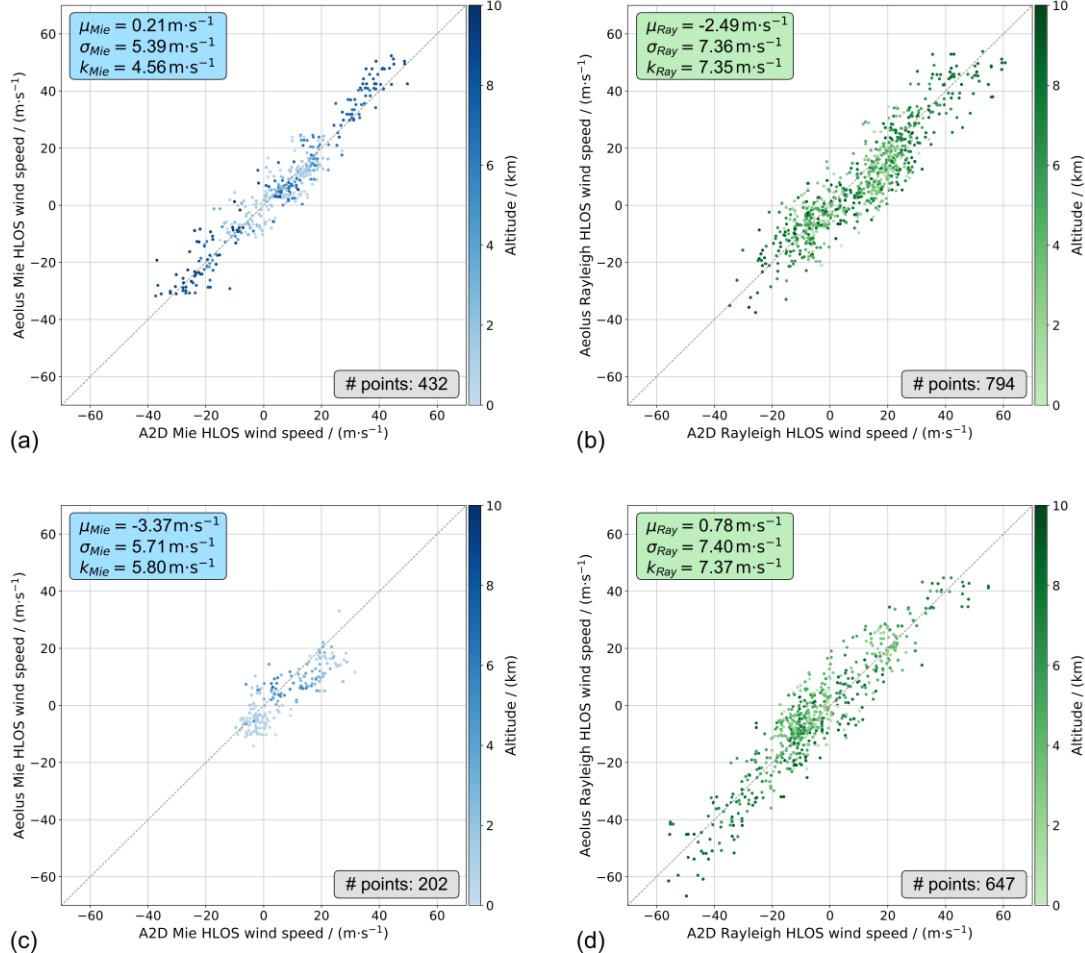

**Figure 12.** Scatterplots comparing the Aeolus L2B Mie-cloudy (a) and Rayleigh-clear winds (b) with the A2D Mie and Rayleigh winds from all underflight legs along ascending orbits during the AVATARI campaign. The bottom panels show the corresponding scatterplots for the Mie (c) and Rayleigh (d) lidar-lidar comparison from all underflight legs along descending orbits of the satellite. The data points are colour-coded with respect to the bottom altitude of the respective bins used for comparison. The mean bias (μ), standard deviation (σ) and scaled median absolute deviation ($k$) are provided in the respective boxes.

The scatterplots in Fig. 12 show a good agreement of the Aeolus and A2D Mie winds with a nearly vanishing bias for ascending orbits (panel (a)), whereas it is -3.4 m·s$^{-1}$ for descending orbits (panel (c)). Considering the slightly negative Mie wind bias of the A2D with respect to the 2-μm DWL (-0.15 m·s$^{-1}$ LOS corresponds to -0.15 m·s$^{-1}$ / sin (20°) = -0.44 m·s$^{-1}$ HLOS), the actual Aeolus Mie wind biases account for -0.2 m·s$^{-1}$ for the ascending and -3.8 m·s$^{-1}$ for the descending orbits, respectively. The number of compared winds (202) and the wind speed range (-10 m·s$^{-1}$ to 30 m·s$^{-1}$) are significantly smaller in the latter case compared to ascending orbits (432; -40 m·s$^{-1}$ to 50 m·s$^{-1}$), as the four underflights mentioned above were either characterized by low- and middle level clouds or even cloud-free conditions (26/09/2019) with only weak or moderate wind speeds. The negative bias of the Aeolus Mie winds for the investigated underflights along descending orbits was confirmed by a comparison





with ECMWF model background data (-1.7 m·s⁻¹), which also showed a significantly smaller bias for the AVATARI dataset along ascending orbits (-0.6 m·s⁻¹). In this context, it should be pointed out that the aforementioned M1 correction aims at reducing the wind bias on a global scale (to below 0.5 m·s⁻¹) which does not eliminate larger biases on regional scales like

those investigated here for the AVATARI campaign. Since the Mie wind bias shows only weak dependence on the telescope temperature, as opposed to the Rayleigh winds, the M1 correction mainly corrects for globally constant offsets (Weiler et al., 2021). The color-coding of the scatters in Fig. 12 which describes the bottom altitude of the bin reveals that the Aeolus Mie winds are positively biased at higher altitudes on ascending orbits. Further analysis suggests that this is due to a wind speed dependence of the Mie bias, potentially due to the application of an imperfect Mie response calibration (Marseille et al., 2021).

The spread between the Aeolus and A2D Mie wind data results from the fact that the random errors of the two lidar instruments approximately add up quadratically according to (Lux et al., 2020a):

$$k_{\text{total}} \approx \sqrt{k_{\text{A2D}}^2 + k_{\text{Aeolus}}^2} \; . \tag{1}$$

Using $k_{\text{A2D}}$ = 1.27 m·s⁻¹ / sin(20°) = 3.71 m·s⁻¹ and $k_{\text{total}}$ = 4.56 m·s⁻¹ (ascending), respectively $k_{\text{total}}$ = 5.80 m·s⁻¹ (descending), one can estimate the actual random errors of the Aeolus L2B Mie winds to be 2.6 m·s⁻¹ (ascending) and 4.4 m·s⁻¹ (descending). Comparison with ECMWF model background data revealed random errors between 3.1 m·s⁻¹ and 3.5 m·s⁻¹ for the two datasets. While this is in fair agreement with the random error derived from the Aeolus-A2D-comparison for ascending orbits given the dissimilar data overlap of the A2D, Aeolus and model winds, the value of 4.4 m·s⁻¹ is significantly larger than that from the

model comparison, suggesting that the respective A2D and Aeolus Mie datasets from the four underflights along descending orbits is too small to be statistically significant.

Considering the Rayleigh channel, the number of compared wind results and the wind speed ranges are more balanced since high wind speeds related to the jet stream were measured with the Aeolus and A2D Rayleigh channels on both descending and ascending orbits. Like for the Mie channel, different biases are evident for the Aeolus Rayleigh winds: It is close to -2.1 m·s⁻¹

for the ascending orbits (-2.49 m·s⁻¹ + 0.13 m·s⁻¹ / sin (20)) and around 1.2 m·s⁻¹ (0.78 m·s⁻¹ + 0.13 m·s⁻¹ / sin (20)) for descending orbits. The more negative bias for ascending orbits is in line with the statistical comparison of the Aeolus Rayleigh winds with the ECMWF model background data, although the discrepancy is much smaller here (-0.7 m·s⁻¹, -0.4 m·s⁻¹). Different Rayleigh wind biases with respect to ECMWF model data for ascending and descending orbits on a global scale were also reported by Martin et al. (2021), although this study relied on the preliminary Aeolus L2B wind data, i.e. without

M1 correction. The results presented here suggest that, despite the implementation of the M1 correction for the reprocessing of the Aeolus wind data, a distinction between ascending and descending orbits is necessary in the error assessment. The random error derived from the Aeolus-A2D-comparison for both data subsets are [(7.4 m·s⁻¹)² –(1.78 m·s⁻¹ / sin (20))²]^{1/2} = 6.2 m·s⁻¹. This result is confirmed by the ECMWF model comparison which yields only slightly smaller values (ascending: 5.8 m·s⁻¹, descending: 5.6 m·s⁻¹). The parameters derived from the statistical comparison of the Aeolus L2B winds against the

A2D and ECMWF model background wind data are summarized in Table 5.





When comparing the derived random errors for the Aeolus winds from the AVATARI campaign to those from other validation studies, it is important to consider the respective range gate settings. For instance, the error assessment performed in the framework of the WindVal III and AVATARE campaigns showed Rayleigh random errors with respect to the 2-µm DWL of about 3.9 m·s⁻¹ and 4.3 m·s⁻¹, respectively (Witschas et al., 2020). During these campaigns, the range bin thickness was 1 km,
whereas the vertical resolution was set to be higher (0.5 km) during the AVATARI campaign. Since this change in the settings involves a reduction in signal level by a factor of two, larger random errors are expected. This is particularly true for the Rayleigh channel where the precision is primarily limited by the Poisson noise and thus by the signal to noise level. In order to verify this effect, the AVATARI winds from the L2B product were post-processed such that the effective vertical resolution was also 1 km. For this purpose, the winds from three adjacent vertical bins (index $i$) were averaged as follows:


$$v_{new} = \frac{1}{4} v_{i-1,\mathrm{orig}} + \frac{1}{2} v_{i,\mathrm{orig}} + \frac{1}{4} v_{i+1,\mathrm{orig}}.$$ (2)

In this way, the wind speed measured in a certain range bin is smoothed with those of the neighbouring vertical bins, while the contribution of the latter to the averaged wind speed is 25% each. At the same time, the number of bins remains unchanged so
that the number of compared winds entering the statistical comparison with the other datasets is also unaffected. The procedure, however, requires that the regarded range bin has two neighbouring bins with valid wind data, which is e.g. not the case for the uppermost and lowermost range bin of every wind profile. Statistical comparison of the modified L2B data with the A2D and the ECMWF model background showed that the change in mean bias that is introduced by the smoothing to 1 km vertical resolution is insignificant, as expected. In contrast, a considerable influence is evident for the Rayleigh random error which is
reduced by approximately a factor of $\sqrt{2}$ from around 6.0 m·s⁻¹ to around 4.2 m·s⁻¹ for both ascending and descending orbits, thus being comparable to the precision of the Aeolus Rayleigh-clear winds determined in previous validation campaigns, but exceeding the mission requirements (2.5 m·s⁻¹ in the troposphere) (ESA, 2016). Concerning the Mie random error, the impact of the bin averaging is much smaller and accounts for only 0.1 m·s⁻¹. One reason is that the precision of the Mie wind speed determination (by the fringe imaging technique) is less dependent on the signal level, as Mie winds are typically obtained from
clouds providing strong backscatter so that the signal level rather depends on the mean optical depth within the range gate. Another aspect is the fact that the L2B Mie wind data coverage is much sparser so that three adjacent vertical bins with valid wind data are only rarely available. Hence, the bin averaging procedure is only applied to much fewer bins compared to the Rayleigh channel.





**Table 5.** Statistical comparison of the Aeolus L2B Mie-cloudy and Aeolus L2B Rayleigh-clear winds against the A2D Mie and Rayleigh as well as the ECMWF model background winds for all underflights performed during the AVATARI campaign. The corresponding scatterplots of the Aeolus-A2D-comparison are shown in Fig. 12. The statistics are derived after adaptation of the A2D and ECMWF model data to the respective L2B measurement grids, i.e., a vertical bin thickness of 500 m. The mean bias and scaled MAD for the Aeolus-A2D-comparison are additionally provided after consideration of the A2D bias and random error with respect to the 2-µm DWL and given in parentheses.

| Statistical parameter | Aeolus Mie-cloudy vs. A2D Mie | | Aeolus Mie-cloudy vs. ECMWF model | | Aeolus Rayleigh-clear vs. A2D Rayleigh | | Aeolus Rayleigh-clear vs. ECMWF model | |
|---|---|---|---|---|---|---|---|---|
| | ascending | descending | ascending | descending | ascending | descending | ascending | descending |
| Number of compared bins | 432 | 202 | 686 | 285 | 794 | 647 | 1263 | 899 |
| Mean bias | 0.2 m·s$^{-1}$ (-0.2 m·s$^{-1}$) | -3.4 m·s$^{-1}$ (-3.8 m·s$^{-1}$) | -0.6 m·s$^{-1}$ | -1.7 m·s$^{-1}$ | -2.5 m·s$^{-1}$ (-2.1 m·s$^{-1}$) | 0.8 m·s$^{-1}$ (1.2 m·s$^{-1}$) | -0.7 m·s$^{-1}$ | -0.4 m·s$^{-1}$ |
| Standard deviation | 5.4 m·s$^{-1}$ | 5.7 m·s$^{-1}$ | 3.9 m·s$^{-1}$ | 4.2 m·s$^{-1}$ | 7.4 m·s$^{-1}$ | 7.4 m·s$^{-1}$ | 6.3 m·s$^{-1}$ | 6.0 m·s$^{-1}$ |
| Scaled MAD | 4.6 m·s$^{-1}$ (2.6 m·s$^{-1}$) | 5.8 m·s$^{-1}$ (4.4 m·s$^{-1}$) | 3.5 m·s$^{-1}$ | 3.1 m·s$^{-1}$ | 7.4 m·s$^{-1}$ (6.2 m·s$^{-1}$) | 7.4 m·s$^{-1}$ (6.2 m·s$^{-1}$) | 5.8 m·s$^{-1}$ | 5.6 m·s$^{-1}$ |

## 5 Summary and conclusions

Owing to its representative design and operating principle the A2D is one of the key instruments for the Cal/Val activities during the Aeolus mission and for the preparation of a potential Aeolus follow-on mission. Therefore, substantial effort has been put into the continuous development of its experimental setup, software implementation and wind retrieval algorithm. In recent years, the focus was on the latter in order to overcome two major deficiencies that have limited the instrument's wind accuracy and precision. A novel QC scheme ensures that only backscatter return signals that were detected during proper co-alignment between the transmit and receive path are further processed in the wind retrieval. This approach diminishes large Rayleigh wind errors that are caused by angular variations, particularly in the near field of the instrument. Moreover, large Mie wind biases related to the skewness of the FI fringe were reduced by averaging the wind speed over adjacent vertical bins containing strong gradients of the particle backscatter signal, e.g., from edges of thick clouds. The retrieval modifications considerably decreased the random error of both channels from more than 2.0 m·s$^{-1}$ to about 1.5 m·s$^{-1}$ (LOS) when comparing the wind datasets from entire campaigns against the high-accuracy 2-µm DWL winds. Furthermore, thanks to the drastic reduction of the systematic Rayleigh wind error in the near field, the measurement range could be extended closer to the aircraft from about 8 km to about 10 km altitude. This enabled the accurate sampling of strong wind gradients and high wind speeds in the area of jet streams during the AVATARI campaign.

Thanks to the improved A2D performance, the Aeolus L2B HLOS winds could be validated with better accuracy. For this purpose, collocated wind measurements from ten underflights in the North Atlantic region around Iceland in September 2019 were compared with each other after harmonization of the two datasets. The statistical analysis yielded different Rayleigh and Mie biases for ascending and descending orbits of Aeolus. While the Mie wind bias nearly vanishes for ascending orbits, it is



-3.8 m·s⁻¹ when only underflights along descending orbits are considered. The comparison against ECMWF model background
winds shows a smaller negative bias of around -1.7 m·s⁻¹, which could be explained by the rather small dataset from the four
underflights along descending orbits lacking statistical significance. The Mie random errors are determined to be around 3 m·s⁻¹
for both campaign data subsets. The L2B Rayleigh winds were found to be negatively biased by about -2.1 m·s⁻¹ along
ascending orbits, whereas a positive bias of around 1.2 m·s⁻¹ is evident on descending orbits. A smaller discrepancy was derived
from the model comparison (-0.7 m·s⁻¹, -0.4 m·s⁻¹) which however also confirmed a Rayleigh random error of about 6.0 m·s⁻¹
for both directions. This value is reduced by about √2 to 4.2 m·s⁻¹ when the vertical bin thickness of 500 m, that was set during
the AVATARI campaign, is increased to 1000 m in post-processing, yielding a random error that is comparable to previous
airborne campaigns. The error assessment shows that, despite the mitigation of the two major error sources realized in the
frame of the reprocessing of the L2B product, the Rayleigh and Mie winds still exhibit biases of up to about 2 m·s⁻¹ (neglecting
the high Aeolus Mie bias relative to the A2D for descending orbits). This is mainly due to the fact that the M1 correction as
part of the new processor aims at diminishing the wind bias on a global scale which however does not exclude larger biases
on regional scales. Nevertheless, the systematic errors are significantly smaller than those determined for the preliminary
WindVal III and AVATARE datasets by means of the A2D (Lux et al., 2020a) and 2-µm DWL (Witschas et al., 2020),
respectively. In the latter study, a Rayleigh wind bias of -4.6 m·s⁻¹ was determined for ascending orbits which is reduced
to -2.1 m·s⁻¹ according to the A2D comparison in the present work.

Despite the increased accuracy and precision of the A2D, the assessment of the Aeolus systematic and random error is primarily
the task of the 2-µm DWL, given its still much higher accuracy of better than 0.1 m·s⁻¹ (Weissmann et al., 2005; Witschas et
al., 2017). The A2D rather serves to study instrument-related issues that help to identify error sources of the satellite instrument
as well as to explore possible measurement techniques and algorithm refinements which cannot be readily tested with the
Aeolus measurements modes and processors due to operational constraints. In this sense, the A2D is an ideal testbed to verify
novel correction schemes as those presented in this study.

Although ALADIN does not suffer from an incomplete telescope overlap within its measurement range and, thanks to its
monostatic design, features an intrinsically stable transmit-receive co-alignment, apart from long term alignment evolutions
(Witschas et al., 2021), the implemented QC based on the horizontal Rayleigh spot positions may be also applicable to the
Aeolus processor. For ALADIN, angular variations of the backscatter signals incident on the Rayleigh channel FPIs primarily
arise from orbital cycles of the temperature distribution across the telescope primary mirror. The resulting variations in the
Rayleigh wind bias are currently corrected by a multiple linear regression model which describes the wind bias, relative to the
ECMWF model, as a function of various temperature sensors located on the primary telescope mirror (Weiler et al., 2021).
Preliminary analysis shows that the horizontal Rayleigh spot position derived from the uppermost atmospheric range bin is
correlated with the orbital position of the satellite. However, further investigation is needed to assess whether the correlation
of this parameter with the Rayleigh wind bias is strong enough such that it can be exploited for an alternative correction scheme
potentially even without the need for ECMWF model winds as a reference. Like for the A2D, the vertical incidence angle,
represented by the vertical spot position, is even more crucial for the systematic error of the Aeolus Rayleigh channel, but not



available during wind mode due to the operation principle of the ACCD. However, as the present study has demonstrated, effective error correction is possible with the one-dimensional information only, supposedly because beam motions mainly occur along a fixed straight (diagonal) line in the plane spanned by the two axes so that the changes in the known horizontal spot position and the unknown vertical spot position are coupled with each other. The applied QC scheme might become even more relevant for Aeolus follow-on mission in the case that bi-static design is chosen with separate transmit and receive paths, thus necessitating an active co-alignment control which might show similar limitations as that of the A2D. The same holds valid for the atmospheric backscatter lidar ATLID (ATmospheric LIDar) onboard ESA's forthcoming Earth Explorer mission EarthCARE (Earth Cloud, Aerosol and Radiation Explorer), scheduled for launch in 2023, which also relies on a bi-static architecture and thus makes use of a co-alignment sensor at receiver chain level for ensuring that the emitted laser beam is aligned with the receiver FOV (do Carmo et al., 2021). However, there are also other receiver designs that may be applicable in future spaceborne Doppler wind lidars, e.g., based on Michelson or Mach–Zehnder interferometers (Herbst and Vrancken, 2016; Tucker et al.; 2018; Baidar et al., 2018; Bruneau and Pelon, 2021), which allow for larger acceptance angles, thus relaxing the requirements on the transmitter–receiver co-alignment.

The correction approach that was applied to A2D Mie winds with large opposing biases in vertically neighbouring range bins might be also transferrable to the Aeolus wind retrieval with a wind gradient threshold adapted to the Aeolus LOS. Although the Mie fringe measured with the ALADIN instrument exhibits a smaller degree of skewness than the A2D Mie fringe, similar error patterns were detected in atmospheric scenes with thick clouds in the lower troposphere. Here, a more detailed analysis of larger Aeolus datasets containing such scenes and those with extraordinarily strong vertical wind speed gradients is necessary to evaluate whether the Aeolus Mie winds are affected by a similar error source and could potentially be corrected by comparable means. Besides ALADIN the presented correction schemes, particularly for the Rayleigh channel, may also be applicable to other DWLs that rely on the utilization of FPIs and that are sensitive to variations of the incidence angle.

The development of the A2D wind retrieval will be further pursued with a focus on the cross-talk between the Rayleigh and Mie channel. In particular, the influence of narrowband Mie backscatter signals on the Rayleigh channel response in case of thin clouds or low atmospheric aerosol loadings is an important aspect that will be addressed in the near future. This topic will also be of high relevance for the Aeolus wind and aerosol products, especially with regard to a recent Aeolus validation campaign that was conducted in the tropics in September 2021.





*Data availability.*

The presented work includes data of the Aeolus mission that is part of the European Space Agency (ESA) Earth Explorer Programme. This includes the reprocessed L2B wind product (Baseline 2B10) for the period of the AVATARI campaign (from 9 September through 29 September, 2019) that is publicly available and can be accessed via the ESA Aeolus Online Dissemination System. The processor development, improvement and product reprocessing preparation are performed by the

Aeolus DISC (Data, Innovation and Science Cluster), which involves DLR, DoRIT, TROPOS, ECMWF, KNMI, CNRS, S&T, ABB and Serco, in close cooperation with the Aeolus PDGS (Payload Data Ground Segment).

*Competing interests.* The authors declare that they have no conflict of interest.

*Author contribution.* OL, CL and OR conducted the A2D wind observations. CL was the PI of the AVATARE and AVATARI campaigns. OL, FW and UM analysed the A2D data and developed the methodology to compare the A2D, 2-µm DWL and Aeolus L2B datasets. BW and SR conducted the 2-µm DWL measurements and processed the 2-µm DWL data. CL, OR, AG and AS conducted the flight planning. The paper was written by OL with contributions from all co-authors.

*Acknowledgements.* We thank ESA colleagues Thorsten Fehr (Aeolus scientific campaign coordinator), Anne Grete Straume (Aeolus mission scientist) and Jonas von Bismarck (Aeolus data quality manager) for their support of the study. The authors are also grateful to the DLR flight experiments department for the realization of the two airborne campaigns within only six months.

*Financial support.* The AVATARE and AVATARI campaigns was supported by the European Space Agency (grant no. 4000128136/19/NL/IA). The first author was partly funded by a young scientist grant by the ESA within the DRAGON 4 program (grant no. 4000121191/17/I-NB).



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
