# Peer review of "Retrieval improvements for the ALADIN Airborne Demonstrator in support of the Aeolus wind product validation"

_Atmospheric Measurement Techniques, 2021_

## Author Comment (AC1)

***Response to Referee Comment (RC1) on***

*Retrieval improvements for the ALADIN Airborne Demonstrator*
*in support of the Aeolus wind product validation*
*(*https://doi.org/10.5194/amt-2021-366*)*

We appreciate the referee's very insightful and helpful remarks on our manuscript. The responses to the individual comments and the corresponding changes that will be made to the manuscript are presented in the following.

General comment:

*This manuscript describes the improvements made to the ALADIN Airborne Demonstrator (A2D) instrument's wind retrieval algorithm for both the Rayleigh and Mie winds. A novel quality control (QC) scheme is implemented to filter Rayleigh wind measurements that are impacted by telescope alignment. This new QC scheme makes some of the data in the near field region (higher altitude) available for the ALADIN wind validation. This is an improvement since all data in the near field region were previously filtered out. Quality of A2D Mie winds are improved by vertically averaging Mie wind results with large bias of opposing sign in adjacent range bins. As a consequence of Fringe skewness, presence of strong scatterers (e.g. clouds) in the range gate overlap region results in winds with opposing sign in those adjacent range bins.*

*The authors very clearly demonstrate the improvement to the retrieved A2D winds with the new retrieval algorithm by comparing them with the concurrent wind measurements made by the 2 um coherent Doppler lidar. Improved applicability of the A2D data for ALADIN winds validation is also shown. These retrieval improvements made to the A2D algorithm also informs potential improvements that could be made for the ALADIN wind retrievals as well as inform future lidar developments especially Doppler lidar using Fabry-Perot and Fizeau interferometers.*

*Overall, the paper is very well written and it should serve as a good reference for future Doppler lidar retrieval algorithm. I recommend this manuscript for publication following the authors addressing my minor comments and edits listed below.*

Comment #1:

*QC scheme for the Rayleigh winds is also applied to the Mie winds. This is stated in the paper (line 399-402) but is buried among all the details. While its effect on the Mie winds comparison might be minimal, it does remove 30-60% of data so, I think the authors need to include a statement in the abstract or summary to highlight this.*

Response to Comment #1:

We agree that this is an important aspect which should be made more clearly. Therefore, the following sentence was added to the summary in the context of the developed QC:

> The QC is likewise applied to sort out the Mie measurements with suboptimal co-alignment, although the Mie channel is less sensitive to alignment variations. Since a large fraction (30 to 60%) of the Rayleigh and Mie measurements is rejected with the current QC settings, a refined alignment QC scheme that acts individually on the Mie channel is foreseen for the future.

Comment #2:

*Please consider adding mean bias (μ), standard deviation (σ) and scaled median absolute deviation (k) symbols to description in text to make it easier for reader to go between figure and text.*

Response to Comment #2:

The following paragraph was added in Sect. 3.2 to introduce the statistical parameters that are used further on in the text:

> The inset in panel (d) includes the A2D systematic wind error with respect to the 2-µm DWL, expressed as the mean bias
>
> $$\mu = \frac{1}{n}\sum_{i=1}^{n}\left(v_{i,\text{A2D}} - v_{i,2-\mu m}\right), \tag{1}$$
>
> i.e., the mean of the wind speed differences that were measured with the two lidars with $n$ being the number of comparable winds after the exclusion of gross errors (see Sect. 3.4.1). The A2D random error is given in terms of the standard deviation
>
> $$\sigma = \sqrt{\frac{1}{n-1}\sum_{i=1}^{n}\left[\left(v_{i,\text{A2D}} - v_{i,2-\mu m}\right) - \mu\right]^2}. \tag{2}$$
>
> In addition to the standard deviation, the scaled median absolute deviation (scaled MAD) is calculated according to
>
> $$k = 1.4826 \cdot \text{median}\left|\left(v_{i,\text{A2D}} - v_{i,2-\mu m}\right) - \text{median}\left(v_{i,\text{A2D}} - v_{i,2-\mu m}\right)\right|. \tag{3}$$
>
> The scaled MAD is a more robust measure of the wind error variability than the standard deviation, as it is more resilient to outliers in the dataset. If the analyzed data are normally distributed, the standard deviation and scaled MAD are identical.

Also, the symbols describing the mean bias (μ), standard deviation (σ) and scaled MAD (k) were added in the text where applicable.

Comment #3:

*There is no details about the ECMWF comparison. Only the results are presented. I think the authors need to provide some details on how ECMWF model data were interpolated for comparison. While it is out of the scope of the paper, have you compared ECMWF and ALADIN winds over the AVATARI region over an extended period to test the significance of the A2D results? Have you compared ECMWF model background winds against 2 um coherent Doppler lidar winds to assess its accuracy? This would be of relevance to results presented in Table 5.*

Response to Comment #3:

We agree that the model comparison is not properly introduced in Sect. 4. We therefore included the following paragraph at the beginning of the section:

> The ECMWF model background winds, i.e., without assimilation of the Aeolus winds, were additionally exploited to compare the Aeolus L2B winds with the model data. For this purpose, the AUX_MET data (meridional and zonal wind component) were averaged onto the L2B grid using the same aerial averaging formalism (Marksteiner, 2013) that was also used for the harmonization of the A2D and L2B datasets. Subsequent projection of the horizontal wind vector from the model onto the Aeolus HLOS vector then allowed for the validation of the L2B wind results using the ECMWF model. The model winds from the AVATARI campaign were, in turn, compared to the 2-µm DWL data to assess its accuracy. The comparison yielded a slightly negative bias of -0.2 m/s and a scaled MAD of 1.7 m/s.

More details on the AUX_MET data are added earlier in the manuscript in the context of the adaptation of the A2D winds to the Aeolus viewing geometry:

> The conversion of the A2D LOS winds to the satellite HLOS involved the use of wind data from the 2-µm DWL (zonal and meridional wind components) and the ECMWF model background that is included in the Aeolus auxiliary meteorological file (AUX_MET). The latter contains vertical profiles of the ECMWF operational short-range forecast model data at 136 pressure levels along the predicted Aeolus track with a horizontal resolution of about 22 km (Rennie et al., 2020).

The Aeolus winds were compared to the ECMWF model background at a global scale and over longer periods (from late 2018 through end of 2019) by Martin et al. (2021). The study also revealed different Rayleigh and Mie biases for ascending and descending orbits that additionally show seasonal variations and also vary with latitude and, to a smaller extent, with longitude.

However, this analysis relies on preliminary Aeolus data without the implementation of the M1 temperature correction, whereas the present study validated the reprocessed Aeolus winds (baseline 2B10) so that a comparison of the validation results is not possible. The effect of the M1 correction scheme on the Aeolus wind results is presented by Weiler et al. (2021b), showing a residual Rayleigh wind bias over the North Atlantic region around Iceland in August 2019 that is slightly negative, thus confirming the findings derived from the Aeolus-A2D-comparison. The significance of the A2D results is additionally strengthened by the comparison of the Aeolus winds against the AVATARI dataset from the 2-µm DWL which revealed systematic and random errors of the Mie and Rayleigh winds that are in good agreement with the A2D results when considering the different data overlap regions of the 2-µm DWL, A2D and Aeolus winds. The statistical comparison of the reprocessed Aeolus wind product against the 2-µm DWL data is not yet published. We therefore refrain from mentioning the results in the present manuscript.

Comment #4:

*Please consider replacing the phrases "vanishing bias", "wind error vanishes". It just seems like bias magically disappeared. Consider "bias decreases to near zero" or "wind error =0".*

Response to Comment #4:

The text was changed accordingly.

Comment #5:

*Figure 6: Please define wind error. Is it the difference compared to 2 um winds? Please clarify.*

Response to Comment #5:

Indeed, the wind speed difference compared to the 2-µm DWL winds is referred to as A2D wind error. In this sense, the mean bias is the mean over all wind speed differences, as defined in Eq. (1). The meaning of the wind error is clarified in the caption of Fig. 6.

Comment #6:

*Figure 5d, e: Consider using a different color scale for greater readability. Something like Fig 5a. Same with Fig 8d, 8e, 10d, 10e, 12.*

Response to Comment #6:

The colour scales in Figs. 5, 8, 10 and 12 were changed to improve the readability.

Comment #7:

*Figure 5e: The lines are grey not green as mentioned in the text.*

Response to Comment #7:

Thanks for noticing. The caption was corrected.

Comment #8:

*Line 217: Just curios what are range gates # 1, 3, and 5 used for.*

Response to Comment #8:

From the 25 range gates (from #0 to #24), three range gates are used for detecting the background light (#0), signals resulting from the voltage at the analogue-to-digital converter (detection chain offset, #2) and the internal reference signal (#4), respectively. The interjacent range gates #1, #3 and #5 act as buffers to avoid leakage into the neighbouring range gates, so that atmospheric backscatter signals are collected in the remaining 19 range gates.

The description of the A2D detector was extended to clarify the use of the range gates #1, #3, and #5. In addition, the slightly different detector readout scheme for ALADIN was added.

> When the A2D is operating in so-called lidar mode, the summed-up signals from 25 images, i.e. 25 rows, are transferred to a memory zone of the ACCD one after another. Each row represents one range gate, from which three are used for detecting the solar background radiation (range gate #0), signals resulting from the voltage at the analogue-to-digital converter (detection chain offset, DCO, range gate #2) and the internal reference signal (range gate #4), respectively. Another three range gates (#1, #3, #5) act as buffers to avoid leakage into the neighbouring range gates, so that 19 range gates are available for collecting the atmospheric backscatter signals (range gates #6 to #24). Note that, for ALADIN, the detection of the internal reference signals does not require a dedicated range gate, as the timing requirements are much more relaxed given the much longer delay between the backscattered return and the emitted pulses. Also, the Aeolus ACCDs make use of so-called "virtual" pixels to determine the DCO instead of using a range gate (Weiler et al., 2021a). Consequently, no buffer range gates are necessary and only one range gate is reserved for detecting the (solar) background which leaves 24 atmospheric range gates for ALADIN.

Comment #9:

*Line 405: Define MAD when first used. Also include how scaled MAD is calculated.*

Response to Comment #9:

This issue was addressed in the response to Comment #2.

Comment #10:

*Line 451-455: Does this mean the new algorithm artificially smears wind in case of strong vertical wind gradients?*

Response to Comment #10:

Indeed, in case of strong vertical wind gradients, a similar "dipole-like" characteristic of the wind bias in neighbouring range gates is observed due to the coarse vertical resolution of the A2D (and ALADIN). However, the systematic errors, or bias differences in the affected bins, are usually smaller compared to those instances where the dipole structure is caused by strong backscatter gradients. Therefore, the wind gradient threshold was set to 5 m·s$^{-1}$·km$^{-1}$ (LOS), corresponding to vertical gradients of the HLOS wind speed of about 15 m·s$^{-1}$·km$^{-1}$. This is above strong gradients that are, e.g., measured below the jet stream (Lux et al., 2018). Using this threshold, bins that are affected by the fringe skewness effect are well identified while leaving bins unflagged that show typical wind gradients with respect to the upper neighbour bin. Nevertheless, since the both effects can occur simultaneously and also enhance each other, e.g., at cloud tops, it cannot be fully excluded that actual wind gradients are artificially smeared in some cases. Proper differentiation between the two effects is challenging and subject of future studies, with the aim to improve both the A2D and Aeolus wind data quality.

The following paragraph was added to Sect. 3.3 to elaborate on this aspect:

The used wind gradient threshold of 5 m·s$^{-1}$·km$^{-1}$ (LOS), corresponding to vertical gradients of the HLOS wind speed of about 15 m·s$^{-1}$·km$^{-1}$, was found to adequately distinguish between cases where the opposing bias is caused by strong vertical backscatter or wind gradients, respectively. Nevertheless, since the two effects can occur simultaneously in rare cases and also enhance each other, e.g., at cloud tops, it cannot be fully excluded that actual wind gradients are artificially smeared by the applied algorithm in some cases.

Comment #11:
*Line 635: delete "both".*
Response to Comment #11:
Done.

---

## Author Comment (AC2)

***Response to Referee Comment (RC2) on***

*Retrieval improvements for the ALADIN Airborne Demonstrator*
*in support of the Aeolus wind product validation*
*(*https://doi.org/10.5194/amt-2021-366*)*

We are grateful for the referee's very valuable and helpful comments on our manuscript which will certainly improve its quality. The responses to the individual comments and questions are presented in the following together with the corresponding changes that will be made for the revised manuscript.

General comment:

*The authors improve the ALADIN Airborne Demonstrator (A2D) Rayleigh and Mie wind data quality by modifications of the A2D wind retrieval algorithm. The authors also demonstrate that the retrieval modifications decrease the bias and random error of both Rayleigh and Mie channels when comparing the wind datasets from AVATARE and AVATARI campaigns against a collocated 2-µm coherent Doppler lidar. The comparison results between A2D Mie and Rayleigh wind data and the Aeolus measurements are of great importance to understand the performance of the reprocessed Aeolus L2B wind products (Baseline 2B10). The paper is well written and should be accepted after the following minor revisions.*

Comment #1:

*Line 73: "FPIs" should be revised to "Fabry–Pérot interferometers (FPIs)".*

Response to Comment #1:

The text was changed accordingly.

Comment #2:

*Tables 1 and 2: Please define the number of A2D observations and the number of Aeolus observations. Are they the numbers of A2D and Aeolus wind profiles used for the comparisons? Is the number of Aeolus Rayleigh wind profiles the same as the number of Aeolus Mie wind profiles? Please clarify.*

Response to Comment #2:

The number of A2D observations in Tables 1 and 2 indeed corresponds to the number of wind profiles along the Aeolus underflight leg which are identical for the A2D Rayleigh and Mie channel. For Aeolus, the situation is more complicated. Here, the number of wind observations defines the number of profiles in the L1B product which is related to the signal acquisition time of 12 s; referred to as

basic repeat cycle (BRC) with a horizontal averaging length of about 90 km. Each BRC consists of 30 measurements in analogy to the A2D signal processing that is described in section 3. The Aeolus wind retrieval as part of the L2B processor, however, involves a so-called grouping algorithm for discriminating so-called Rayleigh-clear and Mie-cloudy winds. Here, groups of measurements are produced irrespective of the original BRCs and hence the resultant L2B wind results may span across subsequent BRCs. This is possible due to the continuous nature of the Aeolus data along the orbit ground-track (Rennie et al., 2020). The grouping algorithm is performed independently for the Mie and Rayleigh channel and results in different horizontal averaging for the wind results. Typically, Mie winds require fewer measurement bins to achieve a given level of precision compared to the Rayleigh winds, with the expected levels of backscatter from e.g. clouds.

Consequently, the number of Aeolus Rayleigh and Mie wind profiles cannot be inferred from the number of wind observations given in Tables 1 and 2. The information in the tables is meant to give an overview on the length of the sampled Aeolus swath via the number of covered Aeolus observations rather than providing detailed data on the amount of validated wind results.

For the revised manuscript, the number of A2D and Aeolus observations provided in Tables 1 and 2 are defined in the text:

> While the number of A2D observations corresponds to the number of wind profiles with a horizontal averaging length of about 3.6 km, one Aeolus observation, also referred to as basic repeat cycle (BRC), is spread over a nominal horizontal averaging length of about 90 km and can contain multiple wind profiles, especially for the Mie channel, as a result of the so-called grouping algorithm which is part of the L2B processor (Rennie et al., 2020).

Additional information on the Aeolus data granularity is provided in section 3 as follows:

> For Aeolus, each observation or BRC takes 12 s and consisted of $P = 19$ pulses and $N = 30$ measurements for the period of the two campaigns in 2019. Since the signals from one pulse are lost during readout of the ACCD ($P - 1$ setting), each BRC contains the signals from 540 pulses, as opposed to the A2D where $(10 - 2) \cdot 70 = 560$ pulses form one observation ($P - 2$ setting).

Comment #3:

*Lines 166-267: In my opinion, the A2D technical details described in Section 3 can be condensed, because you described them in Lux et al. (2020a).*

Although most of the A2D technical details in section 3 are already covered in Lux et al. (2020a), the description also contains several links to the instrument's deficiencies and the methods to tackle them (telescope overlap, alignment sensitivity, Rayleigh spot positions, Mie fringe, range gate overlap). In our opinion, the level of detail is necessary to ensure comprehensibility of the following sections of the manuscript. Therefore, we would like to refrain from shortening the instrument's description.

Comment #4:

*Lines 215-219: The range gate settings are different from those described in Lux et al. (2020a). Please add some further explanations on that.*

Response to Comment #4:

The change in the A2D range gate settings is motivated in section 2 as follows:

> Most importantly, following the recommendations formulated in Lux et al. (2020a), the range gate settings of the A2D were optimized such that a higher number of small and medium-sized range gates were located at altitudes above 4 km at the expense of lower resolution towards the ground. The higher resolution at higher altitudes allowed for a better vertical sampling of high wind speed gradients, e.g., related to jet streams near the tropopause, and hence delivered wind data over a wider wind speed range to be used for the validation of the Aeolus wind product.

In addition, the description of the A2D detector was extended to clarify the use of the range gates #1, #3, and #5. Also, the slightly different detector readout scheme for ALADIN was added.

> When the A2D is operating in so-called lidar mode, the summed-up signals from 25 images, i.e. 25 rows, are transferred to a memory zone of the ACCD one after another. Each row represents one range gate, from which three are used for detecting the solar background radiation (range gate #0), signals resulting from the voltage at the analogue-to-digital converter (detection chain offset, DCO, range gate #2) and the internal reference signal (range gate #4), respectively. Another three range gates (#1, #3, #5) act as buffers to avoid leakage into the neighbouring range gates, so that 19 range gates are available for collecting the atmospheric backscatter signals (range gates #6 to #24). Note that, for ALADIN, the detection of the internal reference signals does not require a dedicated range gate, as the timing requirements are much more relaxed given the much longer delay between the backscattered return and the emitted pulses. Also, the Aeolus ACCDs make use of so-called "virtual" pixels to determine the DCO instead of using a range gate (Weiler et al., 2021a). Consequently, no buffer range gates are

necessary and only one range gate is reserved for detecting the (solar) background which leaves 24 atmospheric range gates for ALADIN.

**Comment #5:**

*Line 274: "detection chain offset (DCO)" should be revised to "DCO" because the DCO is already defined in Line 218.*

**Response to Comment #5:**

The text was changed accordingly.

**Comment #6:**

*Figure 5e: "light green lines" should be revised to "thin black lines". Why no data is seen in altitude ranges of 1.5 to 2.0 and 3.0 to 3.5 km?*

**Response to Comment #6:**

The line colours in Fig. 5e were changed to grey and made thicker for better readability and the caption was adapted accordingly.

The fact that no data is seen in the mentioned altitude ranges is related to the coarse vertical resolution of the A2D where the range gate thickness was set to 1.2 km in the lower troposphere (see Response to Comment #4). As the altitude of the range gate centres vary with the flight altitude, the data points corresponding the lower range gates are spread over several hundreds of meters in Fig. 5e. Nevertheless, the separation of the bin centres by 0.6 km and 1.2 km, depending on the chosen vertical integration time for each range gate, is still visible.

**Comment #7:**

*Lines 356-359: I don't understand that it is sufficient to only consider the range gate #6. It should be better explained.*

**Response to Comment #7:**

The Rayleigh wind error that is caused by alignment fluctuations is correlated among all range gates, since variations in the FPI incidence angle influence the Rayleigh spectrometer response independent of the vertical integration setting. However, the impact is largest for range gate #6, i.e., in the instrument's near field, where the backscatter signal is reduced due to the incomplete telescope overlap and the contribution from the large angle parts of the field is most pronounced. This also becomes obvious from Fig. 6, which shows the wind error for each range gate depending on the Rayleigh spot position in range gate #6. At the optimum spot position, not only the wind error for

range gate #6 becomes close to zero, but also the wind error for most of the other range gates. The slope is, however, steepest for the wind measured in range gate #6 which suggests using this data for the determination of the optimum spot position.

In the revised manuscript this aspect is elaborated as follows:

> The Rayleigh wind error that is caused by alignment fluctuations is correlated among all range gates, since variations in the FPI incidence angle influence the Rayleigh spectrometer response independent of the vertical integration setting. This becomes obvious from Fig. 6, where the error is also close to zero for those wind results that were measured in the other range gates at times when the spot position in range gate #6 is close to the determined optimum. Therefore, it is sufficient to only consider the most sensitive range gate #6 for the relationship between spot position and wind error.

Comment #8:

*Lines 359-362: Why do the authors only consider the position of spot B? If the sensitivities of two bandpass filters (A and B) are different, that affects determining the Doppler frequency shift in the Rayleigh channel. It should be better explained.*

Response to Comment #8:

Variations of the incidence angle on the two FPI band pass filters lead to Rayleigh spot motions that are indeed differently large for filters A and B. However, for geometrical reasons, the horizontal spot motions of the two filters are strongly correlated to each other, e.g., both spots move to the periphery or to the centre of the Rayleigh ACCD symmetrically, albeit by a different amount. Consequently, despite the different sensitivities of the two filters to alignment variations, considering the spot motions of only one of the two filters is sufficient for evaluating its relationship to the wind error. Since the position of spot B is more sensitive to alignment variations, mainly due to the fact that filter B is passed after filter A and the beam has travelled a longer path before being incident on the ACCD, it is used for the QC routine.

This aspect is explained in more detail in the revised manuscript:

> For geometrical reasons, the horizontal spot motions of the two filters are strongly correlated to each other, e.g., both spots move to the periphery or to the centre of the Rayleigh ACCD symmetrically. The reason why spot B is preferred over spot A is the higher sensitivity of its position to angular variations which is mainly due to the fact that filter B is passed after filter A and the beam has thus travelled a longer path before being incident on the ACCD. The

higher alignment sensitivity of spot B was confirmed by analysing the correlation between spot position in range gate #6 and the A2D wind error for all Aeolus underflights from the AVATARE and AVATARI campaign.

Comment #9:

*Lines 407-408: Please define mean bias, standard deviation, and scaled mean absolute deviation. "MAD" should be revised to "mean absolute deviation (MAD)".*

Response to Comment #9:

The following paragraph was added in Sect. 3.2 to introduce the statistical parameters that are used further on in the text:

The inset in panel (d) includes the A2D systematic wind error with respect to the 2-µm DWL, expressed as the mean bias

$$\mu = \frac{1}{n}\sum_{i=1}^{n}\left(v_{i,\text{A2D}} - v_{i,2-\mu\text{m}}\right), \tag{1}$$

i.e., the mean of the wind speed differences that were measured with the two lidars with $n$ being the number of comparable winds after the exclusion of gross errors (see Sect. 3.4.1). The A2D random error is given in terms of the standard deviation

$$\sigma = \sqrt{\frac{1}{n-1}\sum_{i=1}^{n}\left[\left(v_{i,\text{A2D}} - v_{i,2-\mu\text{m}}\right) - \mu\right]^{2}}. \tag{2}$$

In addition to the standard deviation, the scaled median absolute deviation (scaled MAD) is calculated according to

$$k = 1.4826 \cdot \text{median}\left|\left(v_{i,\text{A2D}} - v_{i,2-\mu\text{m}}\right) - \text{median}\left(v_{i,\text{A2D}} - v_{i,2-\mu\text{m}}\right)\right|. \tag{3}$$

The scaled MAD is a more robust measure of the wind error variability than the standard deviation, as it is more resilient to outliers in the dataset. If the analyzed data are normally distributed, the standard deviation and scaled MAD are identical.

Also, the symbols describing the mean bias ($\mu$), standard deviation ($\sigma$) and scaled MAD ($k$) were added in the text where applicable.

Comment #10:

*Line 511: "mean absolute deviation (MAD)" should be revised to "MAD" because the MAD is already defined in Line 408.*

Response to Comment #10:

The text was changed accordingly.

Comment #11:

*Section 4: Why do the authors present the results of comparison between Aeolus winds and ECMWF model background winds? Is it out of the scope of the paper? Please add some further explanations on that.*

Response to Comment #11:

The results of the comparison between Aeolus winds and the ECMWF model background winds are included in the manuscript to strengthen the results from the Aeolus-A2D-comparison, particularly regarding the different biases for ascending and descending orbits, and with regard to the residual systematic errors after the first data reprocessing, i.e., the implementation of the M1 correction. We agree that the model comparison is not properly introduced in Sect. 4. We therefore included the following paragraph at the beginning of the section:

> The ECMWF model background winds, i.e., without assimilation of the Aeolus winds, were additionally exploited to compare the Aeolus L2B winds with the model data. For this purpose, the AUX_MET data (meridional and zonal wind component) were averaged onto the L2B grid using the same aerial averaging formalism (Marksteiner, 2013) that was also used for the harmonization of the A2D and L2B datasets. Subsequent projection of the horizontal wind vector from the model onto the Aeolus HLOS vector then allowed for the validation of the L2B wind results using the ECMWF model. The model winds from the AVATARI campaign were, in turn, compared to the 2-µm DWL data to assess its accuracy. The comparison yielded a slightly negative bias of -0.2 m/s and a scaled MAD of 1.7 m/s.

More details on the AUX_MET data are added earlier in the manuscript in the context of the adaptation of the A2D winds to the Aeolus viewing geometry:

> The conversion of the A2D LOS winds to the satellite HLOS involved the use of wind data from the 2-µm DWL (zonal and meridional wind components) and the ECMWF model background that is included in the Aeolus auxiliary meteorological file (AUX_MET). The latter contains vertical profiles of the ECMWF operational short-range forecast model data at 136 pressure levels along the predicted Aeolus track with a horizontal resolution of about 22 km (Rennie et al., 2020).

The Aeolus winds were also compared to the ECMWF model background at a global scale and over longer periods (from late 2018 through end of 2019) by Martin et al. (2021). In accordance with the present work, the study revealed different Rayleigh and Mie biases for ascending and

descending orbits that additionally show seasonal variations and also vary with latitude and, to a smaller extent, with longitude. However, this analysis relies on preliminary Aeolus data without the implementation of the M1 temperature correction, whereas the present study validated the reprocessed Aeolus winds (baseline 2B10) so that a comparison of the validation results is not possible. The effect of the M1 correction scheme on the Aeolus wind results is presented by Weiler et al. (2021b), showing a residual Rayleigh wind bias over the North Atlantic region around Iceland in August 2019 that is slightly negative, thus confirming the findings derived from the Aeolus-A2D-comparison. The significance of the A2D results is additionally strengthened by the comparison of the Aeolus winds against the AVATARI dataset from the 2-µm DWL which revealed systematic and random errors of the Mie and Rayleigh winds that are in good agreement with the A2D results when considering the different data overlap regions of the 2-µm DWL, A2D and Aeolus winds. The statistical comparison of the reprocessed Aeolus wind product against the 2-µm DWL data is not yet published. We therefore refrain from mentioning the results in the present manuscript.

Comment #12:

Technical corrections: Line 685: $\sin(20)$ -> $\sin(20°)$; line 692: $\sin(20)$ -> $\sin(20°)$

Response to Comment #12:

Thanks for noticing. The corrections were made in the text.